# Performance of doubled haploid maize (*Zea mays L.*) testcross hybrids under optimal and drought-stressed environments

Goshime Muluneh Mekasha[1,2]*, Zerihun Demrew Yigezu[1], Adefris Teklewold[3], Manje Gowda[4], Juan Burgueño[5], Yoseph Beyene[4]

1 Hawassa University College of Agriculture/School of Plant and Horticultural Science/Plant Biotechnology, Hawassa, Ethiopia, 2 Ethiopian Institute of Agriculture Research, Hawassa National Maize Research, Hawassa, Ethiopia, 3 International Maize and Wheat Improvement Center, ILRI Campus, Sholla, Addis Ababa, Ethiopia, 4 International Maize and Wheat Improvement Center (CIMMYT), Nairobi, Kenya, 5 Data Science Group, International Maize and Wheat Improvement Center (CIMMYT), México D.F., México

* m.goshime87@gmail.com

## Abstract

Maize (*Zea mays L.*) productivity in Sub-Saharan Africa is increasingly constrained by recurrent drought linked to climate change. Improving yield stability under contrasting moisture conditions remains challenging, especially when breeding materials are derived from parental lines within the same heterotic group (e.g., Group A × Group A), where genetic divergence is limited. We hypothesized that doubled haploid (DH) lines derived from biparental populations still harbor sufficient within-population genetic variation to generate exploitable phenotypic diversity and adaptive differentiation. Furthermore, crossing each DH line with a single-cross tester from the opposite heterotic group provides an effective framework to capture this variation through hybrid performance. In the present work therefore, 855 DH testcross hybrids and six commercial checks were evaluated under optimal and managed drought conditions using an alpha-lattice design. Drought stress was imposed two weeks before flowering until harvest. The mean grain yield under optimal conditions were between 3.66 to 10.36 t ha⁻¹, while 0.16 to 6.13 t ha⁻¹ under drought condition. The mean of top 10 hybrids (≈1.2% selection intensity) outperformed the mean of commercial checks by 60% under optimal conditions and 161% under drought stress. Under optimum conditions, 629 and 286 hybrids exhibited higher heterosis over the mean of checks and the best check, respectively. Similarly, 536 and 235 hybrids surpassed the mean of checks and the best check, respectively, in terms of standard heterosis. The mean and best check yields under drought stress were 1.97 and 3.10 while, 5.91 and 7.09 t ha⁻¹ under optimum condition, respectively. The multi-trait clustering grouped the hybrids into four distinct adaptive categories. Cluster 3, defined primarily by grain yield, integrated temperate introgression with elite tropical backgrounds consistently expressed superior yield performance along with desirable secondary traits. These results

**Data availability statement:** The relevant data are within the manuscript.

**Funding:** Bill and Melinda Gates Foundation (B&MGF), and the United States Agency for International Development (USAID) through the Stress Tolerant Maize for Africa (STMA, B & MGF Grant # OPP1134248).

**Competing interests:** No.

demonstrate that DH lines derived from within-group parental crosses can generate functional diversity and predictable adaptive clusters. Thus supports a cluster-based selection strategy to improve drought tolerance, yield potential, and adaptation in maize breeding programs.

---

## 1 Introduction

Drought is a major constraint to maize production and a significant threat to global food security [1,2]. In Sub-Saharan Africa, the challenge is increasingly driven by the rising frequency of mid-season drought episodes, particularly around flowering, which are often simulated as managed drought in breeding programs to mimic field conditions and identify tolerant genotypes [3,4]. As global temperatures rise, drought intensity and variability are expected to increase, further exacerbating yield instability [1]. At the same time, the global population is projected to reach nearly 10 billion by 2050, requiring a 60–70% increase in food production [5,6].

Maize (*Zea mays* L.) plays a central role in global agriculture as a source of food, feed, and industrial raw materials [7–10]. In eastern and southern Africa, it is the dominant staple crop for rural households and a cornerstone of food security [11]. However, maize productivity in Africa remains low (<2.1 t ha$^{-1}$) compared to the global average (~5.8 t ha$^{-1}$) [8,12], largely due to climate-induced stresses. Among these, drought is the most critical, causing yield losses ranging from 23% to 70%, particularly when it occurs near flowering [13,14]. In the face of climate change, rapidly developing maize germplasm with resistance to key abiotic and biotic stresses is crucial for the resilience of Africa's maize-based cropping systems [15]. The International Maize and Wheat Improvement Center (CIMMYT), in collaboration with various institutions, has successfully developed and deployed stress-tolerant maize cultivars in sub-Saharan Africa, Asia, and Latin America through extensive stress screening and on-farm testing. To further enhance genetic gains in tropical maize breeding, integrating doubled haploid, high-throughput phenotyping, and genomic-assisted breeding with effective data management is essential [16].

Plant breeding remains a key strategy for improving drought tolerance and yield stability. Recent advances such as DH technology enable the rapid development of completely homozygous lines within 1–2 generations, compared to 6–8 generations required in conventional inbreeding, thereby accelerating genetic gain and varietal development [17,18]. This is particularly important for responding to climate variability and shortening breeding cycles. Genetic diversity is fundamental for effective crop improvement [19]. Understanding diversity within breeding materials supports the identification of superior parental lines and the formation of heterotic groups, which are essential for hybrid development [20–22]. However, most hybrid breeding strategies rely on crosses between distinct heterotic groups (e.g., Group A × Group B) to exploit heterosis, while crosses within the same heterotic group are generally considered to have limited genetic divergence and, therefore, lower potential for hybrid vigor [23,24].

Despite this assumption, the extent to which populations derived from within-group crosses can generate useful phenotypic diversity, particularly under drought conditions, remains insufficiently explored [24,25]. This represents a critical knowledge gap in maize breeding, especially when breeding programs are constrained by limited heterotic pools. We therefore hypothesized that DH lines derived from within-heterotic group crosses (e.g., Group A × Group A) maintain sufficient genetic variation to generate exploitable phenotypic diversity and adaptive differentiation under contrasting moisture conditions, and that this variation can be effectively revealed through testcrossing with a single-cross tester from the opposite heterotic group. Accordingly, the present study aimed to evaluate the phenotypic performance of DH maize testcross hybrids under optimal and managed drought conditions, with the objective of identifying superior hybrids and assessing their adaptive patterns across environments.

## 2 Materials and methods

The maize (*Zea mays L.*) DH lines were developed from 16 single-cross and three bulk populations formed using marker-assisted recurrent selection techniques. Details of the base population characteristics and the genetic sources used to develop the 834 DH lines are presented in Table 1. These lines were generated through in vivo haploid induction following the standard CIMMYT DH induction procedure [17]. Each of the 855 DH lines was crossed with a single cross-tester from the opposite heterotic group, yielding 855 hybrids. This work did not require specific permission. One of the mandates of the International Maize and Wheat Improvement Center (CIMMYT) is to develop new lines and hybrids and evaluate them across different biotic and abiotic stresses to identify the best-adapted hybrids for commercialization across various parts of sub-Saharan Africa.

**Table 1. List of maize single cross-source populations, pedigree, characteristics, and number of DH lines used in the study.**

| Code | Pedigree | HG | Characteristic | Lines Phenotype | Tester |
|------|----------|----|----------------|-----------------|--------|
| pop1 | CKDHL0500/CML202 | B | Drought tolerant by high-yield | 49 | CML312/CML442 |
| pop 2 | CKDHL0089/CML494 | B | High yield potential by MLN tolerant | 50 | CML312/CML442 |
| pop 3 | CKDHL0089/CML543 | B | High yield potential by High yield potential | 48 | CML312/CML442 |
| pop 4 | CKDHL0089/CKDHL120918 | B | High yield potential by MLN tolerant | 49 | CML312/CML442 |
| pop 5 | CKLTI0139/CKDHL120918 | A | Temperate introgression by MLN-tolerant | 49 | CML312/CML442 |
| pop 6 | CKDHL0228/CKLTI0133 | A | Elite line by temperate introgression | 50 | CML395/CML444 |
| pop7 | CKLTI0133/CKDHL120312 | A | Temperate introgression by MLN-tolerant | 48 | CML395/CML444 |
| pop 8 | CKLTI0227/CML543 | B | Temperate introgression by high-yield potential | 48 | CML312/CML442 |
| pop 9 | CML395/CKLTI0052 | B | Drought tolerant by Temperate introgression | 49 | CML312/CML442 |
| pop 10 | CKLTI0272/CKDHL0228 | A | Temperate introgression by elite line | 49 | CML395/CML444 |
| pop 11 | CKLTI0344/CKLTI0147 | A | Temperate introgression by Temperate introgression | 50 | CML395/CML444 |
| pop 12 | CML395/CKLMARSI0002 | B | drought tolerant by drought tolerant | 48 | CML312/CML442 |
| pop 13 | CKLTI0200/CML442 | A | Temperate introgression by drought-tolerant | 47 | CML395/CML444 |
| pop 14 | CKLTI0200/CKLTI0036 | A | Temperate introgression by Temperate introgression | 53 | CML395/CML444 |
| pop 15 | CKLTI0272/CML442 | A | Temperate introgressed line by drought-tolerant | 50 | CML395/CML444 |
| pop 16 | CKLTI0368/CKDHL0228 | A | Temperate introgressed line by high-yielding line | 47 | CML395/CML444 |
| pop 17 | DTMAMARSKEN5Bulk | A | Population developed through RCGS | 2 | CML395/CML444 |
| pop 18 | DTMA MARS KEN6 Bulk | A | Population developed through RCGS | 4 | CML395/CML444 |
| pop 19 | DTMA MARS KEN7 Bulk | A | Drouth tolerant by drought tolerant | 44 | CML395/CML444 |
| Mixed | | | unknown | 21 | |
| Total | | | | 855 | |

HG = Heterotic Group; letters A and B represent Heterotic Group A and Heterotic Group B, respectively. 'Pop' followed by a number indicates the population number.

## 2.1 Test crosses performance evaluation

The 855 newly developed hybrids were organized into 17 trial sets connected through a common check strategy according to Lado et al. [26], enabling valid comparisons across sets. Each trial included four to six shared commercial check hybrids, which served as references to account for environmental and design differences among sets. In total, 861 hybrids (including checks) were evaluated using an alpha-lattice design with two replications. Field evaluations were conducted in Kenya during the 2019 growing season under two well-watered environments and one managed drought stress environment, allowing all hybrids to be assessed on a common performance scale across trials. Drought stress was imposed by withholding irrigation approximately two weeks before flowering and maintaining water deficit conditions until harvest, allowing progressive soil moisture depletion. In contrast, well-watered trials received supplemental irrigation whenever moisture stress was observed, ensuring near-optimal soil moisture throughout the growing period. Although the exact amount of water applied was not quantified in millimeters, the drought treatment was characterized. The experiments for optimal management were conducted at two locations in Kenya: Kakamega and Kiboko, while moisture stress experiments were performed solely at Kiboko. Kakamega is situated at 34°46'E and 0°17'N, with an elevation of 1,585 meters above sea level (masl). On the other hand, Kiboko is located at 37°44'E and 2°13'S, with an elevation of 975 masl.

Each plot consisted of a single row measuring 5 m in length, with a spacing of 0.75 m between rows and 0.75 m between plants. Initially, two seeds per hill were planted, and after three weeks, plants were thinned to one per hill, achieving a density of 53,333 plants per hectare. Fertilizer was applied at 60 kg of nitrogen and 60 kg of phosphorus per hectare, with nitrogen split into applications at planting and six weeks later. Weeds were controlled by hand weeding.

Plant aspect (PAS) and ear aspect (EAS), which are breeder-assessed traits, were evaluated using a visual scoring system. Both traits were rated on a 1–5 scale, where 1 indicates excellent and 5 indicates poor performance. Plant aspect (PAS) was assessed based on overall plant appearance, including vigor, uniformity, plant type, and absence of disease or lodging. Ear aspect (EAS) was evaluated considering ear characteristics such as size, uniformity, filling, and freedom from pest or disease damage. Scoring was conducted at physiological maturity across all plots to ensure consistency in evaluation for PAS and during harvesting time for EAS. The data recorded for each plot included days to 50% silking and anthesis, anthesis-silking interval (days), plant and ear heights measured from the base to the first tassel branch and from the base to the node bearing upper ear, respectively, root and stalk lodging percentages (plants leaning more than 30 degrees from vertical, and plants broken at or below the highest ear node, respectively) were also assessed. In optimum and stressed conditions, grain moisture was measured after harvesting and shelling the ears. For well-watered conditions, ear weight was used to estimate grain yield, adjusted to 12.5% moisture content.

## 2.2 Phenotypic data analysis

Before data analysis, the anthesis-silking interval (ASI) was normalized using $\ln\sqrt{(ASI + 10)}$ as suggested by Bolafios and Edmeades [27]. The analysis of variance for all traits was done separately for each location and combined across locations using DeltaGen online software applying mixed model procedures [28]. Trait BLUPs were computed using DeltaGen online software (https://www.deltagen.agr.nz) [28]. Genotypes, replication, and blocks within replications were considered random effects for individual location analysis. In combined analysis, location was considered as fixed effects whereas genotypes, replication within location, blocks within replication and locations, and genotypes by location interaction were considered random effects. Under both individual and combined analysis, the variance was partitioned into relevant sources of variation to test for the difference among genotypes and genotypes by environment interaction. The following model was used for individual locations.

$$y_{jkb} = \mu + L_j + r_k + i(r)_b + e_{jkb}$$

where $y_{jkb}$ represents the phenotypic trait analysis on the k$^{th}$ replicate of the $j^{th}$ test cross in the $b^{th}$ incomplete block, $\mu$ is an intercept (Grand Mean), $L_j$ is the random effect of the $j^{th}$ maize test cross, $r_k$ is the random effect of the $k^{th}$ replicate that is independently, identically, and normally distributed (iid), ib(r)$_b$ denotes the random effect of the $b^{th}$ incomplete block within the $k^{th}$ replicate $\sigma2\,ib(r)$ being the variance of the incomplete block within each replicate, and $e_{jkb}$ is error variance. For multi-location data, the following GLM-mixed model will be used:

$$Y_{ijrk} \ = \ \mu \ + \ L_j \ + \ Rr\,(L_j) \ + \ Bk\,[Rr\,(L_j)] \ + \ G_i \ + \ GL_{ij} \ + \ \varepsilon_{ijrk}$$

where $Y_{ijrk}$ is the mean value of genotype i at location j in replicate r within the block k; μ is the general mean; $L_j$ is the fixed effect of the location $j$; $Rr\,(L_j)$ is the random effect of the replicate r within location $j$; $Bk\,[Rr\,(L_j)]$ is the random effect of the incomplete block $k$ within replicate r and location $j$; $G_i$ is the random effect of genotype i; $GL_{ij}$ is the random and effect of the genotype × location interaction; and $\varepsilon_{ijrk}$ is the random residual error assumed independent and identically normally distributed with mean zero and variance $\sigma^2\varepsilon$. Standard heterosis (SH) of new genotypes over the best check and mean of the checks (SH) was calculated as SH = 100 × (F1–best check)/best check, and SH = 100 × (F1– mean of check)/mean of checks, respectively. Genotypes exceeding the respective critical difference (CD) values were considered significantly superior to the checks.

Narrow-sense heritability (h$^2$) was estimated as the ratio of additive genetic variance to total phenotypic variance (h$^2$ = V_A/V_P), representing the proportion of phenotypic variation attributable to additive effects. In this study, additive genetic variance was approximated as one-quarter of the among-family variance (¼σ²_A), following standard half-sib family assumptions [28]. This measure excludes non-additive components such as dominance and epistasis. Genetic variance was estimated from 855 new doubled haploid (DH) crosses evaluated as half-sib families alongside six standard checks. Heritability estimates for combined analyses were based on entry means across environments and replications, while estimates for individual locations were based on entry means across replications. The contribution of each source of variation to total variability was computed using R software (Version 4.4.1) [29].

## 2.3 Cluster analysis

For clustering analysis, both single-trait and multivariate approaches were employed. Single-trait clustering, particularly for grain yield, used mean BLUEs (Best Linear Unbiased Estimates) from each location as input in the DeltaGen online tool [28], enabling identification of genotypes with consistent performance across environments. In contrast, multivariate analysis incorporated combined data across locations and management conditions (optimum and drought), allowing simultaneous evaluation of multiple traits and broader phenotypic variation. K-means clustering was then applied in R to group genotypes based on overall similarity, minimizing within-cluster variation through iterative reassignment of cluster centroids. This approach produced clusters of genotypes with similar performance patterns, facilitating selection and diversity assessment across environments.

## 2.4 Genotype ranking and population mean

Mean grain yield (GY; t ha$^{-1}$) and percentage yield advantage relative to the mean of the check varieties were calculated for the top 20 genotypes from each breeding population evaluated across three environments (Kakamega_OPT, Kiboko_OPT, and Kiboko_DT). Genotypes were ranked based on mean best linear unbiased estimates (BLUEs) across environments. Population means were calculated as the arithmetic mean of the mean BLUEs of the top 20 genotypes within each population, except for Population 17, Population 18, and the check group, for which the mean was calculated using all contributing genotypes. Accordingly, pop 17 and pop 18 were represented by two and four genotypes, respectively, while the check group included six commercial varieties.

                                                                    

# 3 Results

## 3.1 Additive genetic variance component and narrow-sense heritability

The variance components, narrow-sense heritability estimates, and their corresponding standard errors (in brackets) are presented in Table 2. Under optimal management at Kiboko, narrow-sense heritability for grain yield (GY) was moderate (0.33). In contrast, anthesis date (AD) showed a higher heritability (0.54), while plant height (PH) and ear height (EH) each had heritability estimates of 0.59. The result showed that heritability estimates for phenological and plant architectural traits (AD, PH, and EH) were consistently higher than those for GY across environments, indicating greater genetic control and stability for these traits. Under moisture stress management at Kiboko, narrow-sense heritability for GY was 0.46. The heritability of AD was 0.65, whereas PH and EH each exhibited heritability estimates of 0.63.

The coefficient of variation (CV) for grain yield was highest under the Kiboko moisture-stressed condition (37.73%) compared to the optimal environments, indicating substantially greater relative variability under drought stress (Table 2). This elevated CV is expected under high-intensity drought conditions (just before two weeks of flowering and onward), particularly in drought-prone environments such as Kiboko, where water limitation amplifies differences among genotypes. Under stress, small differences in drought tolerance mechanisms such as root architecture, stomatal regulation, water-use efficiency, problem of synchrony of male and female flowering which enforce limited fertility can lead to large differences in grain yield, thereby increasing variability relative to the mean. Furthermore, drought stress typically reduces the overall mean yield (2.38 t ha$^{-1}$ in this case), which inherently inflates the CV since it is expressed as a percentage of the mean. The combination of reduced mean performance and heterogeneous genotype responses to stress contributes to the observed high CV.

The present work also revealed the presence of substantial variation in genotypes across various conditions and locations. Among genotypes, significant variance was observed for traits such as GY, AD, ear position height (EPH), ear aspect score (EAS), PH, EH, silking date (SD), and anthesis-silking interval (ASI) at a high significance level ($\alpha = 0.01$). Differences for ear per plant (EPP) and plant aspect score (PAS) were significant at $\alpha = 0.05$, while no significant variance was noted for grain moisture content (MOI). The interaction between genotype and location was highly significant for most traits, including GY, AD, EPH, EAS, PAS, MOI, PH, EH, and SD, highlighting the varying performance of genotypes across different locations. However, this interaction was not significant for EPP and ASI.

Location-specific results further underscore the variability. At Kakamega, Kiboko, and Kiboko_DT, the genotypic variance was highly significant for traits such as GY, AD, EPH, EAS, MOI, PH, EH, EPP and SD. Variance in PAS was significant at Kakamega and Kiboko under optimal conditions.

In combined analysis, blocking effects within replication and location were notably significant for GY, MOI, PH, EH, and SD, indicating the importance of experimental design with inclusion of block. Under optimal conditions, blocking effects were significant for GY, MOI, PH, and EH at Kakamega and Kiboko, and for AD at Kiboko. The blocking effect was also significant for EAS and PAS at Kakamega. Furthermore, the blocking effects under moisture stressed condition, at Kiboko, were significant for GY, AD, EPP, MOI, PH, and EH. These results confirmed the importance of considering both genotype and environmental conditions in breeding programs to optimize yield performance across diverse settings.

## 3.2 Variance based on sum square contribution

For traits such as GY, PH, and EH, the percent sum square contribution is notably higher under water-stressed conditions by 49, 62, and 44%, respectively, indicating that genotype variance for these traits was more pronounced when plants were subjected to water stress (Table 3). Similarly, for the ASI and other phenological parameters, the percent sum square contribution was significantly higher under water stress, ranging from 55 to 93%, further emphasizing the impact of stress on genotype variance for these traits (Table 3). Even though the variance for genotype × location interaction was significant for most traits, the proportion of total variability explained by this interaction relative to that of genotype was low to

**Table 2. Additive variance component and narrow sense heritability estimates in 855 test crosses and 6 checks evaluated under water stress and optimal rain-fed environments in Kenya.**

| Test environment | Source of variation | GY | AD | EPH | EPP | EAS | PAS |
|---|---|---|---|---|---|---|---|
| Combined (optimal) | Genotype | 0.39(0.07)*** | 1.98(0.20)*** | 0.0006(0.000)*** | 0.0014(0.0006)* | 0.05(0.01)*** | 0.02(0.009)* |
| | Genotype: Location | 0.63(0.08)*** | 1.69(0.18)*** | 0.0004(0.000)*** | 0.0003(0.001) | 0.06(0.01)*** | 0.08(0.01)*** |
| | Location: REP: BLK | 0.12(0.03)*** | 0.04(0.03)* | 0.0000(0.000)* | 0.0000(0.0001) | 0.003(0.002)* | 0.004(0.002) |
| | Location: REP | 0.01ns | 0.003(0.05) | 0.0000(0.000)** | 0.0001(0.0001) | 0.0006(0.0009) | 0.00(0.000)*** |
| | Residual | 1.86(0.06) | 3.50(11) | 0.0006(0.000) | 0.029(0.001) | 0.26(0.008) | 0.25(0.008) |
| | Heritability | 0.34(0.05) | 0.54(0.03) | 0.62(0.03) | 0.16(0.06) | 0.36(0.04) | 0.14(0.07) |
| | Mean | 6.63 | 65.74 | 0.50 | 1.00 | 2.88 | 2.72 |
| | CV | 20.54 | 2.85 | 4.96 | 16.85 | 17.59 | 18.32 |
| Kakamega_ (Optimal) | Genotype | 1.33(0.13)*** | 4.09(0.31)*** | 0.001(0.0)*** | 0.001(0.001)* | 0.18(0.02)*** | 0.15(0.01)*** |
| | REP: BLK | 0.14(0.05)*** | 0.0(0.0) | 0.00(0.0) | 0.0001(0.0002) | 0.006(0.004)* | 0.01(0.004)*** |
| | REP | 0.02(0.03) | 0.01(0.02) | 0.00(0.0)* | 0.0002(0.0002) | 0.001(0.002) | 0.0(0.001) |
| | Residual | 2.30(0.10) | 4.26(0.19) | 0.001(0.0) | 0.03(0.001) | 0.36(0.02) | 0.26(0.01) |
| | Heritability | 0.37(0.03) | 0.49(0.03) | 0.54(0.02) | 0.05(0.03) | 0.34(0.03) | 0.36(0.03) |
| | Mean | 7.06 | 73.38 | 0.49 | 1.03 | 2.97 | 2.75 |
| | CV | 21.34 | 2.81 | 5.48 | 16.52 | 20.21 | 18.36 |
| Kiboko (Optimal) | Genotype | 0.72(0.08)*** | 3.20(0.23)*** | 0.0011(0.0)*** | 0.003(0.001)** | 0.05(0.006)*** | 0.03(0.009)*** |
| | REP: BLK | 0.08(0.03)*** | 0.14(0.06)*** | 0.0(0.0) | 0.00(0.0) | 0.0004(0.001) | 0.0(0.0) |
| | REP | 0.00(0.00) | 0.00(0.00) | 0.0(0.0) | 0.00(0.0) | 0.0(0.0) | 0.0(0.0) |
| | Residual | 1.43(0.07) | 2.71(0.12) | 0.001(0.0) | 0.03(0.001) | 0.15(0.007) | 0.24(0.01) |
| | Heritability | 0.33(0.03) | 0.54(0.02) | 0.69(0.02) | 0.09(0.03) | 0.24(0.03) | 0.13(0.03) |
| | Mean | 6.18 | 58.11 | 0.51 | 0.97 | 2.78 | 2.65 |
| | CV | 19.3 | 2.84 | 4.38 | 16.99 | 13.92 | 18.24 |
| Kiboko (Moisture stressed) | Genotype | 0.69(0.06)*** | 4.26(0.27)*** | 0.001(0)*** | 0.02(0.002)*** | 0.07(0.01)*** | No data |
| | REP: BLK | 0.06(0.02)*** | 0.07(0.04)* | 0.0(0.0) | 0.0006(0.0004)* | 0.0001(0.002) | |
| | REP | 0.01(0.04)ns | 0.0003(0.005) | 0.0(0.0) | 0.001(0.0008)** | 0.00(0.00) | |
| | Residual | 0.80(0.04) | 2.33(0.11) | 0.001(0.0) | 0.04(0.002) | 0.27(0.01) | |
| | Heritability | 0.46(0.03) | 0.65(0.02) | 0.57(0.02) | 0.34 0.03() | 0.21(0.03) | |
| | Mean | 2.38 | 67.48 | 0.54 | 0.71 | 3.01 | |
| | CV | 37.73 | 2.26 | 4.35 | 26.16 | 17.34 | |
| Test environment | Source of variation | PH | EH | SD | ASI | MOI | |
| Combined | Genotype | 80.33(10.60) *** | 62.53(6.23)*** | 2.33(0.22)*** | 0.0012(0.0004)*** | 0.18(0.13) | |
| | Genotype: Location | 135.96(11.10) *** | 66.37(5.51)*** | 1.49(0.18)*** | 0.0006(0.0005) | 1.69(0.19)*** | |
| | Location: REP: BLK | 14.17(3.46) *** | 6.65(1.69)*** | 0.10(0.04)*** | 0.0002(0.0001) | 0.13(0.05)*** | |
| | Location: REP | 0.36 (1.45) ns | 0.00(0.000) | 0.00(0.000)ns | 0.0002(0.0001)** | 0.08(0.05)** | |
| | Residual | 162.17 | 82.97(2.65) | 4.04(0.13) | 0.02(0.0006) | 4.01(0.13) | |
| | Heritability | 0.43 (0.04) | 0.54(0.03) | 0.57(0.03) | 0.18(0.05) | 0.09(0.06) | |
| | Mean | 247.31 | 124.94 | 66.15 | 2.33 | 20.06 | |
| | CV | 5.15 | 7.29 | 3.04 | 6.16 | 9.98 | |

*(Continued)*

**Table 2.** (Continued)

| Test environment | Source of variation | GY | AD | EPH | EPP | EAS | PAS |
|---|---|---|---|---|---|---|---|
| Kakamega | Genotype | 210.43(15.17)*** | 158.58(10.32)*** | 4.10(0.31)*** | 0.0004(0.0005) | 1.25(0.22)*** | |
| | REP: BLK | 10.45(4.39)*** | 6.12(2.43)*** | 0.05(0.04) | 0.0002(0.0002) | 0.08(0.05)* | |
| | REP | 2.13(2.92) | 0.0(0.0) | 0.0(0.0) | 0.0004(0.0003)* | 0.18(0.14)** | |
| | Residual | 178.16(8.00) | 96.63(4.35) | 4.25(0.19) | 0.0322(0.0012) | 5.14(0.23) | |
| | Heritability | 0.54(0.02) | 0.62(0.02) | 0.49(0.03) | 0.01(0.01) | 0.20(0.03) | |
| | Mean | 263.98 | 131.97 | 73.71 | 0.398 | 22.37 | |
| | CV | 5.05 | 7.44 | 2.8 | 7.73 | 10.14 | |
| Kiboko_OPT | Genotype | 226.09(14.83)*** | 101.35(6.72)*** | 3.47(0.28)*** | 0.0039(0.0)*** | 2.46(0.2)*** | |
| | REP: BLK | 13.47(4.72)*** | 5.55(1.97)*** | 0.19(0.08)*** | 0.0(0.0) | 0.17(0.07)*** | |
| | REP | 0.00(0.004) | 0.15(0.97) | 0.0(0.0) | 0.0(0.0) | 0.0(0.0) | |
| | Residual | 145.32(6.56) | 68.74(3.10) | 3.85(0.17) | 0.008(0.0) | 2.9 | |
| | Heritability | 0.61(0.02) | 0.59(0.02) | 0.47(0.03) | 0.32(0.03) | 0.46(0.03) | |
| | Mean | 230.42 | 117.84 | 58.58 | 2.34 | 17.74 | |
| | CV | 5.23 | 7.03 | 3.35 | 3.88 | 9.59 | |
| Kiboko_DT | Genotype | 230.47(16.71)*** | 138.84(8.83)*** | 6.21(0.47)*** | 0.01(0.001)*** | 4.36(0.52)*** | |
| | REP: BLK | 20.20(6.66)*** | 6.16(2.24)*** | 0.04(0.05) | 0.0002(0.0002) | 0.83(0.29)*** | |
| | REP | 0.001(0.07) | 0.00(0.00) | 0.00(0.00) | 0.000(0.0000) | 0.0(0.0) | |
| | Residual | 203.87(9.20) | 80.51(3.63) | 6.08(0.27) | 0.02(0.0009) | 9.48(0.44) | |
| | Heritability | 0.53(0.02) | 0.63(0.02) | 0.52(0.03) | 0.34(0.03) | 0.32(0.03) | |
| | Mean | 213.74 | 115.78 | 70.37 | 2.54 | 19.92 | |
| | CV | 6.67 | 7.74 | 3.5 | 5.57 | 15.33 | |

Where, GY = Garin yield t/ha, AD = Anthesis date, EPH, Ear position height, EPP = Ear Per Plant, EAS = Ear aspect, PAS = Plant aspect, MOI = Grain moisture content, PH = Plant height, EH = Ear height, SD = Silking date, ASI = Anthesis Silking Interval, REP = Replication, BLK = Block.

**Table 3. Source of variance and their contribution to each trait under the different conditions.**

| Source of variation | DF | GY | SSdif | AD | SSdif | PH | SSdif | EH | SSdif | SD | SSdif | ASI | SSdif | ASI_N | SSdif |
|---|---|---|---|---|---|---|---|---|---|---|---|---|---|---|---|
| | | Combined across well-watered | | | | | | | | | | | | | |
| Genotype | 860 | 36 | −49 | 6 | −93 | 28 | −62 | 44 | −44 | 6 | −91 | 29 | −55 | 27 | −58 |
| Location | 1 | 6 | – | 89 | – | 41 | – | 18 | – | 88 | – | 0 | – | 1 | – |
| Genotype: Location | 860 | 25 | – | 3 | – | 17 | – | 20 | – | 3 | – | 22 | – | 22 | – |
| Location: REP | 2 | 0 | – | 0 | – | 0 | – | 0 | – | 0 | – | 0 | – | 1 | – |
| Location: REP: BLK | 62 | 0 | – | 0 | – | 0 | – | 0 | – | 0 | – | 0 | – | 0 | – |
| Residuals | 2030 | 33 | 22 | 3 | −84 | 14 | −43 | 18 | −8 | 4 | −87 | 49 | 38 | 49 | 45 |
| Total % SS | | 100 | – | 100 | – | 100 | – | 100 | – | 100 | – | 100 | – | 100 | – |
| | | Water stress at Kiboko | | | | | | | | | | | | | |
| Genotype | 860 | 71 | – | 81 | – | 72 | – | 79 | – | 73 | – | 63 | – | 65 | – |
| REP | 1 | 0 | – | 0 | – | 0 | – | 0 | – | 0 | – | 0 | – | 0 | – |
| REP: BLK | 31 | 2 | – | 1 | – | 3 | – | 2 | – | 1 | – | 1 | – | 1 | – |
| Residuals | 985 | 27 | – | 18 | – | 25 | – | 19 | – | 27 | – | 35 | – | 34 | – |
| Total % SS | | 100 | – | 100 | – | 100 | – | 100 | – | 100 | – | 100 | – | 100 | – |

Where, GY = Garin yield t/ha, AD = Anthesis date, PH = Plant height, EH = Ear height, SD = Silking date, ASI = Anthesis Silking Interval, ASI_N = Anthesis Silking Interval normalized, SSdif = sum square difference, df = degree freedom, SS = Sum Square.

moderate across all traits. This finding aligns with the report by Govindaraj et al. [30] who showed that genotype × environment interactions are significant and their contribution to the total variability is often less than the variability attributed to genotype alone.

Conversely, the residual variance showed different patterns. For GY and ASI, residual variance was higher under well-watered conditions by 22 and 38%, respectively, when compared with water-stressed conditions (Table 3). This suggests that while genotype variance was more pronounced under stress, the residual variance, or unexplained variance, was higher in well-watered environments. For PH and EH, the sum square contribution was greater under water stress, by 43 and 8%, respectively, as compared to well-watered conditions (Table 3). Notably, the residual variance for AD and SD was significantly higher under water-stressed conditions by 84 and 87%, respectively, as compared to well-watered conditions, indicating that the unexplained variability for these phenological traits were increased under stressed condition (Table 3). The findings showed that while water stress amplifies the genetic differences for various traits, it also increases the unexplained variance for certain parameters.

### 3.3 Means and ranges under well-watered and drought-stressed conditions

Under moisture-stressed conditions at Kiboko, the mean grain yield was 2.38 t/ha with a heritability estimate of 0.46. The moderate heritability indicates substantial genetic variability among genotypes, suggesting good prospects for selection under drought stress. Although experimental variability was relatively high, the presence of heritable variation supports effective genetic improvement in stress environments (Table 4). Grain yield under moisture-stressed condition was drastically lower by 179.65%, than in non-stressed environments. Water stress also led to reduction in plant height by 15.64% and ear height (EH) by 7.88%, while increasing the anthesis-silking interval (ASI) by 86.14% (Table 5). However, moisture content did not significantly differ between the two conditions (Table 5).

**Table 4. Characteristics of the experimental locations, average yield, narrow sense heritability and CV for 855 test crosses and 6 checks.**

| Location | Year | Type of environment | Number of replications | Plot size(m2) | Plant density/ha | GY(t/ha) | Heritability | CV (%) |
|---|---|---|---|---|---|---|---|---|
| Combined | 2019 | Well-watered | 2 | 3.75 | 53,333 | 6.63 | 0.34 | 20.54 |
| Kakamega | 2019 | Well-watered | 2 | 3.75 | 53,333 | 7.06 | 0.37 | 21.34 |
| Kiboko_OPT | 2019 | Well-watered | 2 | 3.75 | 53,333 | 6.18 | 0.33 | 19.3 |
| Kiboko_DT | 2019 | Drought-stress | 2 | 3.75 | 53,333 | 2.38 | 0.46 | 37.73 |

Where, OPT = well-watered or optimum; DT = drought; ha = hectare; GY = grain yield; T = ton; CV = coefficient of variation; m² = square meter.

**Table 5. Trait BLUE mean performance of test crosses and checks evaluated in drought stress and well-watered environments in Kenya in 2019.**

| Management | GY | AD | EPH | EAS | MOI | PH | EH | SD | ASI | NP |
|---|---|---|---|---|---|---|---|---|---|---|
| Well-watered (Combined) | 6.63 | 65.75 | 0.51 | 2.88 | 20.05 | 247.31 | 124.94 | 66.15 | 0.40 | 17.90 |
| Moisture- stressed | 2.38 | 67.48 | 0.54 | 3.01 | 20.08 | 213.93 | 115.81 | 70.37 | 2.89 | 17.83 |
| Mean Difference | 4.27 | −1.73 | −0.04 | −0.14 | −0.03 | 33.46 | 9.14 | −4.21 | −2.49 | 0.08 |
| Mean Difference in % | 179.65 | −2.57 | −6.64 | −4.50 | −0.15 | 15.64 | 7.88 | −5.99 | −86.14 | 0.47 |
| T-Test (Significance) | ** | ** | ** | ** | ns | ** | ** | ** | ** | ** |

Where, GY = Garin yield t/ha, AD = Anthesis date, EPH, Ear position height, MOI = Grain moisture content, PH = Plant height, EH = Ear height, SD = Silking date, ASI = Anthesis Silking Interval, NP = Number of Plant per plot.

In well-watered environments, GY of the test crosses ranged from 3.66 to 10.36 t/ha, with a mean of 6.63 t/ha. The top ten test crosses exceeded the best check, PH30G19, by 2.10–3.27 t/ha (Table 6). Overall, the mean yield of all test crosses was 12% higher than the checks, and the top ten DH crosses showed a 60% yield advantage. Anthesis date (AD) ranged from 59.48 to 71.70 (mean 65.75 days), while silking days ranged from 59.89 to 72.11 days, with a short mean ASI of 0.4 days. Plant aspect (PAS) (1.74–4.01; mean 2.72) and ear aspect (1.74–3.75; mean 2.88) indicated generally good plant and ear quality. Plant height (PH) ranged from 209.45 to 283.74 cm (mean 247.31 cm), and ear height from 96.16 to 155.90 cm (mean 124.94 cm). Ears per plant (EPP) varied from 0.65 to 1.48, with a mean of 1.0, indicating limited prolificacy under well-watered conditions (Table 6).

Under water stress, the grain yields of test crosses were ranged from 0.16 to 6.13 t/ha (Table 7). The top ten test crosses yield were 1.57 and 3.03 t/ha which was more than the best check (DK777) (Table 7). On average, the new DH test crosses exceeded the mean grain yield of the checks by 21%, and the top ten performing DH test crosses showed a remarkable 161% yield increase over the mean of the checks (Table 7). The performance metrics for the tested genotypes revealed notable variations. The mean for AD ranged from 60.49 to 75.10, with an overall average of 67.48 days, while the SD varied from 62.45 to 80.95 days. The ASI showed a range from −1.44 to 13.97 days, resulting in an overall mean of 2.89 days. For EAS, the overall mean was 3.01, with a range of 1.0 to 5.0. The mean PH ranged from 163.58 to 265.20 cm, averaging 213.93 cm. EH had a mean value of 115.79 cm, with a range from 75.15 to 153.94 cm. These results

**Table 6. The BLUE mean performance of the top ten new testcross and six check hybrids evaluated at two well-watered environments in Kenya.**

| Genotype | Code | GY | AD | SD | ASI | PH | EH | EPH | EPP | SL | EAS | PAS | MOI | NP |
|---|---|---|---|---|---|---|---|---|---|---|---|---|---|---|
| CKDHL1702100 | G135 | 10.36 | 68.04 | 69.62 | 1.53 | 274.17 | 141.89 | 0.52 | 1.10 | 12.86 | 2.75 | 1.92 | 20.75 | 17.19 |
| CKLTI0368 | G849 | 9.91 | 65.05 | 64.18 | −0.93 | 252.48 | 131.78 | 0.52 | 1.02 | 12.33 | 2.26 | 1.74 | 23.30 | 18.18 |
| CKDHL1701770 | G116 | 9.53 | 66.30 | 67.85 | 1.52 | 275.15 | 150.06 | 0.55 | 1.13 | 12.88 | 2.76 | 1.98 | 19.41 | 17.99 |
| CKDHL1717071 | G596 | 9.42 | 67.19 | 68.04 | 0.93 | 253.51 | 130.89 | 0.52 | 1.10 | 12.35 | 2.00 | 2.48 | 22.27 | 19.44 |
| CKDHL1701760 | G113 | 9.39 | 67.06 | 66.35 | −0.76 | 257.12 | 121.02 | 0.47 | 1.13 | 12.51 | 2.02 | 1.95 | 19.68 | 17.22 |
| CKDHL1715861 | G583 | 9.32 | 63.46 | 64.14 | 0.73 | 256.42 | 133.15 | 0.52 | 0.95 | 12.19 | 2.26 | 1.99 | 20.31 | 19.58 |
| CKDHL1715875 | G585 | 9.25 | 64.25 | 64.18 | −0.03 | 249.24 | 125.82 | 0.51 | 1.10 | 12.21 | 2.49 | 2.27 | 19.38 | 17.49 |
| CKDHL1701853 | G123 | 9.23 | 67.15 | 67.05 | −0.01 | 280.23 | 148.13 | 0.53 | 1.10 | 12.61 | 2.75 | 2.41 | 20.42 | 17.23 |
| CKDHL1702333 | G150 | 9.21 | 68.52 | 69.57 | 1.03 | 274.96 | 147.12 | 0.53 | 1.05 | 13.65 | 3.00 | 1.97 | 18.23 | 19.24 |
| CKDHL1717047 | G593 | 9.19 | 65.98 | 66.24 | 0.28 | 232.50 | 120.82 | 0.52 | 1.08 | 12.46 | 2.51 | 2.49 | 21.25 | 18.01 |
| DK 777 | G856 | 6.32 | 65.64 | 65.94 | 0.28 | 239.06 | 113.57 | 0.48 | 0.99 | 12.03 | 2.30 | 2.46 | 20.54 | 18.10 |
| Duma 43 | G857 | 5.63 | 61.35 | 62.64 | 1.27 | 244.77 | 111.63 | 0.46 | 1.00 | 12.34 | 3.00 | 2.80 | 18.48 | 17.80 |
| H516 | G858 | 5.74 | 65.45 | 67.47 | 2.01 | 268.76 | 144.31 | 0.54 | 1.03 | 13.08 | 3.13 | 2.95 | 19.35 | 16.20 |
| PAN 4M19 | G859 | 4.36 | 59.52 | 61.31 | 1.76 | 243.87 | 113.97 | 0.47 | 0.98 | 12.58 | 3.24 | 3.47 | 17.99 | 16.69 |
| PH 30G19 | G860 | 7.09 | 64.67 | 66.21 | 1.53 | 253.35 | 118.12 | 0.47 | 1.02 | 12.08 | 2.39 | 2.50 | 21.44 | 18.27 |
| WH505 | G861 | 6.29 | 66.42 | 66.88 | 0.47 | 252.14 | 129.14 | 0.51 | 0.92 | 12.25 | 2.97 | 2.75 | 20.12 | 17.49 |
| Mean of all | | 6.63 | 65.75 | 66.15 | 0.40 | 247.31 | 124.94 | 0.51 | 1.00 | 12.58 | 2.88 | 2.72 | 20.06 | 17.90 |
| Mean of crosses | | 6.63 | 65.76 | 66.15 | 0.39 | 247.29 | 124.96 | 0.51 | 1.00 | 12.58 | 2.88 | 2.72 | 20.07 | 17.90 |
| Mean of checks | | 5.91 | 63.84 | 65.07 | 1.22 | 250.33 | 121.79 | 0.49 | 0.99 | 12.39 | 2.84 | 2.82 | 19.65 | 17.42 |
| Minimum value | | 3.66 | 59.48 | 59.89 | −2.03 | 209.45 | 96.16 | 0.43 | 0.65 | 11.88 | 1.74 | 1.74 | 15.62 | 13.09 |
| Maximum value | | 10.36 | 71.70 | 72.11 | 4.75 | 283.74 | 155.90 | 0.62 | 1.48 | 14.06 | 3.75 | 4.01 | 23.96 | 20.34 |
| LSD | | 2.45 | 3.63 | 3.67 | 1.99 | 28.91 | 20.39 | 0.05 | 0.24 | 23.48 | 0.85 | 0.89 | 3.77 | 3.29 |

Where GY = Grain yield, AD = Anthesis date, SD = Silking date, ASI = Anthesis Silking interval, PH = Plant Height, EH = Ear Height, EPH = Ear Position Height, EPP = Ear per plant, SL = Stalk Lodging, EAS = Ear Aspect, PAS = Plant Aspect, MOI = Grain Moisture, NP = Number of Plant per plot, LSD = List Significant Difference.

**Table 7. Mean (BLUE) mean performance of the top 10 new testcross and six check hybrids evaluated under water-stressed condition.**

| Genotype | Code | GY | AD | SD | ASI | PH | EH | EPH | EPP | ER | RL | SL | EAS | MOI | NP |
|---|---|---|---|---|---|---|---|---|---|---|---|---|---|---|---|
| CKDHL1715839 | G576 | 6.13 | 66.14 | 67.89 | 1.49 | 234.10 | 128.88 | 0.55 | 0.90 | 7.70 | 0.00 | 19.06 | 1.99 | 22.99 | 15.07 |
| CKDHL1715313 | G500 | 5.80 | 68.40 | 69.45 | 0.97 | 247.74 | 134.75 | 0.54 | 1.11 | 0.00 | 0.00 | 37.47 | 2.50 | 21.50 | 12.93 |
| CKDHL1720436 | G792 | 5.25 | 68.74 | 72.04 | 3.46 | 240.00 | 142.11 | 0.59 | 0.91 | 16.25 | 0.00 | 40.75 | 3.00 | 22.70 | 20.43 |
| CKDHL1718483 | G707 | 5.16 | 68.03 | 69.44 | 1.41 | 237.66 | 132.95 | 0.56 | 1.00 | 6.25 | 0.00 | 36.39 | 3.00 | 22.01 | 16.03 |
| CKDHL1717281 | G643 | 5.14 | 67.61 | 69.08 | 1.55 | 228.27 | 132.99 | 0.58 | NA | NA | 0.00 | 20.23 | 2.01 | 21.12 | 17.58 |
| CKDHL1715563 | G533 | 4.99 | 67.50 | 69.05 | 1.57 | 238.97 | 126.82 | 0.53 | 0.90 | 0.00 | 0.00 | 26.17 | 1.50 | 21.04 | 17.98 |
| CKDHL1717501 | G684 | 4.84 | 68.05 | 71.00 | 2.97 | 213.15 | 119.37 | 0.56 | NA | NA | 0.00 | 53.85 | 3.01 | 20.65 | 18.95 |
| CKDHL1718481 | G706 | 4.80 | 64.86 | 66.94 | 1.98 | 219.82 | 114.11 | 0.52 | 0.94 | 13.35 | 0.00 | 41.00 | 3.00 | 21.64 | 18.62 |
| CKDHL1707536 | G375 | 4.72 | 63.64 | 66.99 | 3.43 | 216.56 | 117.20 | 0.54 | 1.00 | 13.35 | NA | 18.77 | 2.99 | 17.95 | 16.44 |
| CKLTI0272 | G847 | 4.67 | 64.42 | 66.45 | 1.98 | 237.48 | 122.59 | 0.52 | 1.00 | 0.00 | 0.00 | 9.71 | 2.00 | 21.82 | 18.03 |
| DK 777 | G856 | 3.10 | 67.25 | 69.13 | 1.85 | 208.04 | 108.39 | 0.52 | 0.81 | 2.24 | 0.00 | 10.70 | 2.09 | 20.17 | 18.09 |
| Duma 43 | G857 | 1.97 | 63.19 | 68.24 | 5.07 | 213.01 | 103.23 | 0.48 | 0.74 | 12.29 | 0.33 | 16.85 | 3.00 | 18.58 | 17.49 |
| H516 | G858 | 1.42 | 69.28 | 72.72 | 3.45 | 215.52 | 127.70 | 0.60 | 0.59 | 14.03 | 0.00 | 37.35 | 3.12 | 19.64 | 13.87 |
| PAN 4M19 | G859 | 1.12 | 63.03 | 70.08 | 7.08 | 181.69 | 103.22 | 0.56 | 0.60 | 8.35 | 0.00 | 17.12 | 3.00 | 20.16 | 19.92 |
| PH 30G19 | G860 | 2.11 | 65.85 | 71.05 | 5.18 | 218.92 | 113.43 | 0.52 | 0.66 | 5.83 | 0.18 | 12.35 | 2.73 | 20.63 | 18.42 |
| WH505 | G861 | 2.11 | 68.37 | 71.09 | 2.71 | 214.20 | 117.74 | 0.55 | 0.57 | 9.06 | 0.00 | 15.81 | 3.23 | 19.78 | 16.76 |
| Mean of all | | 2.38 | 67.48 | 70.37 | 2.89 | 213.93 | 115.79 | 0.54 | 0.71 | 7.76 | 0.16 | 23.40 | 3.01 | 20.08 | 17.83 |
| Mean of crosses | | 2.38 | 67.49 | 70.37 | 2.88 | 213.78 | 115.81 | 0.54 | 0.71 | 7.75 | 0.16 | 23.43 | 3.02 | 19.92 | 17.83 |
| Mean of checks | | 1.97 | 66.16 | 70.38 | 4.22 | 208.56 | 112.28 | 0.54 | 0.66 | 8.63 | 0.09 | 18.36 | 2.86 | 19.83 | 17.42 |
| Minimum value | | 0.16 | 60.49 | 62.45 | −1.44 | 163.58 | 75.15 | 0.44 | 0.09 | −0.24 | 0.00 | −0.19 | 1.00 | 12.31 | 9.87 |
| Maximum value | | 6.13 | 75.10 | 80.95 | 13.97 | 265.20 | 153.94 | 0.63 | 1.46 | 66.65 | 41.65 | 85.38 | 5.0 | 30.44 | 21.65 |
| LSD | | 1.78 | 3.01 | 4.85 | 3.88 | 28.18 | 17.71 | 0.05 | 0.37 | 18.30 | 4.35 | 33.96 | 1.03 | 6.09 | 4.76 |

Where GY = Grain yield, AD = Anthesis date, SD = Silking date, ASI = Anthesis Silking interval, PH = Plant Height, EH = Ear Height, EPH = Ear Position Height, EPP = Ear Per Plant, RL = Root Lodging, SL = Stalk Lodging, EAS = Ear Aspect, ER = Ear Rot, MOI = Grain Moisture, NP = Number of Plant per plot, LSD = List Significant Difference.

indicate that the genotypes exhibit significant variation in performance under water stress conditions, particularly in terms of growth metrics.

### 3.4 Standard heterosis for grain yield

Under optimum condition significant variation in standard heterosis for grain yield was observed among genotypes when evaluated over the best check (PH30G19) and the mean of checks. Several genotypes expressed positive and statistically significant heterosis, indicating superior yield performance relative to the checks. Genotype G135 recorded the highest heterosis over the best check (46.18%) and over the mean of checks (75.48%) (Table 8). Other genotypes including G849, G596, G113, G583, G585, and G123 also exhibited significant positive heterosis over both standards heterosis. Heterosis values ranged from −48.32% to 46.18% over the best check and from −37.96% to 75.48% over the mean of checks, reflecting wide genetic variability for grain yield.

Under drought conditions, standard heterosis for GY over the best check (DK777) and the mean of checks varied widely among genotypes and showed high levels of statistical significance. Several genotypes demonstrated pronounced positive heterosis, indicating strong yield superiority under moisture stress. Genotype G576 exhibited the highest heterosis over the best check (97.71%) and over the mean of checks (210.97%), both highly significant. Genotypes G500, G792, G707, G643, and G533 also showed substantial and significant positive heterosis over both standards. The

**Table 8. Standard heterosis (%) of the top 20 maize genotypes for grain yield, along with a summary of genotypes exhibiting positive and negative heterosis under OPT and drought conditions, estimated relative to the best check and the mean of checks.**

| Genotype | Combined data under OPT (best check) | Combined data under OPT (mean of checks) | Genotype | Under drought (best check) | Under drought (mean of checks) |
|---|---|---|---|---|---|
| G135 | 46.18* | 75.48** | G576 | 97.71*** | 210.97*** |
| G849 | 39.76* | 67.77** | G500 | 87.05** | 194.21*** |
| G116 | 34.36 | 61.29** | G792 | 69.47* | 166.55*** |
| G596 | 32.81 | 59.42* | G707 | 66.40* | 161.72*** |
| G113 | 32.45 | 58.99* | G643 | 65.82* | 160.81*** |
| G583 | 31.40 | 57.73* | G533 | 60.89* | 153.05** |
| G585 | 30.47 | 56.62* | G684 | 56.05 | 145.44** |
| G123 | 30.16 | 56.24* | G706 | 54.82 | 143.51** |
| G150 | 29.85 | 55.88* | G375 | 52.26 | 139.48** |
| G593 | 29.60 | 55.58* | G847 | 50.74 | 137.09** |
| G130 | 29.53 | 55.49* | G532 | 50.34 | 136.46** |
| G551 | 29.37 | 55.30* | G339 | 49.61 | 135.32** |
| G613 | 28.96 | 54.80* | G718 | 48.58 | 133.69** |
| G621 | 28.82 | 54.63* | G690 | 46.92 | 131.10** |
| G144 | 28.65 | 54.43* | G376 | 44.67 | 127.54** |
| G617 | 28.11 | 53.78* | G520 | 44.63 | 127.49** |
| G285 | 28.01 | 53.66* | G525 | 44.46 | 127.22** |
| G581 | 27.51 | 53.10* | G545 | 44.39 | 127.11** |
| G628 | 27.44 | 52.99* | G508 | 43.53 | 125.75** |
| G588 | 26.98 | 52.43* | G651 | 43.43 | 125.59** |
| Minimum | −48.32 | −37.96 | | −105.21 | −108.2 |
| Maximum | 46.18 | 75.48 | | 97.71 | 210.97 |
| CD, 0.05 | 2.71 | 2.71 | | 1.74 | 1.74 |
| CD, 0.01 | 3.56 | 3.56 | | 2.29 | 2.29 |
| CD, 0.001 | 4.55 | 4.55 | | 2.93 | 2.93 |
| Number of genotypes with positive or negative standard heterosis | | | | | |
| *_positive | 6 | 94 | | 2 | 39 |
| ns_positive | 230 | 442 | | 285 | 590 |
| ns_negative | 458 | 316 | | 552 | 232 |
| *_negative | 165 | 8 | | 17 | 0 |

Where *_positive, *_negative, ns_positive, and ns_negative indicate genotypes with significant or non-significant positive or negative standard heterosis, respectively (= significant at P ≤ 0.05, 0.01, or 0.001; ns = non-significant). The table also shows the number of such genotypes and the corresponding critical differences (CD) at the 0.05, 0.01, and 0.001 probability levels.

heterosis estimates ranged from −105.21% to 97.71% over the best check and from −108.20% to 210.97% over the mean of checks (Table 8), indicating considerable genetic variation among genotypes for GY expression. Several genotypes surpassed the critical difference thresholds, confirming their statistical superiority.

### 3.5 Grouping of genotypes using GY BLUEs estimate

Cluster and pattern analyses based on grain yield classified the 861 genotypes into three distinct clusters: C1 (239 genotypes), C2 (294 genotypes), and C3 (328 genotypes) (Fig 1). The dendrograms revealed clear relationships

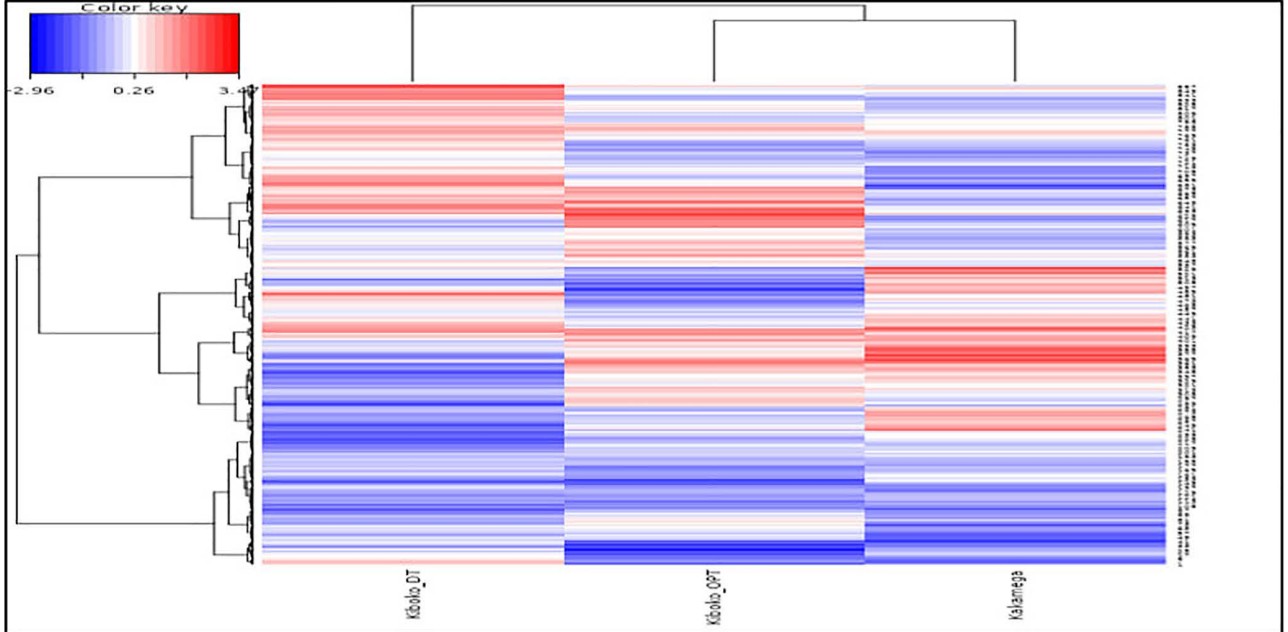

**Fig 1. Hierarchical clustering (dendrogram) of maize genotypes and test locations based on phenotypic grain yield data.** Rows represent individual genotypes, while columns represent the three environments (Kiboko_DT, Kiboko_OPT, and Kakamega). The dendrograms illustrate similarity patterns among genotypes and locations derived from standardized grain yield values. Color intensity reflects relative grain yield performance, where red indicates higher-than-average yield, blue indicates lower-than-average yield, and white represents values close to the overall mean, as shown in the color key. This heatmap highlights genotype-by-environment differentiation and reveals contrasting yield responses of genotypes across drought-stressed, optimal, and high-rainfall environments.

among genotypes and test environments based on yield performance. The two optimal environments, Kiboko_OPT and Kakamega_OPT, clustered closely together, indicating similar genotype responses. In contrast, Kiboko_DT was clearly separated, highlighting the strong impact of imposed drought stress and the resulting phenotypic variation among genotypes. The heatmap further showed that genotypes with good performance under Kiboko_DT were mainly grouped in Clusters 2 and 3, while high-performing genotypes at Kiboko_OPT were also concentrated in Clusters 2 and 3. Superior performance at Kakamega was predominantly associated with Cluster 2.

Clustering based solely on grain yield revealed clear and consistent differences in mean performance across environments. These differences closely matched the population composition within each cluster (Table 9). The yield-based clusters showed distinct response patterns across Kiboko_DT, Kiboko_OPT, and Kakamega_OPT, indicating a strong association between cluster mean performance and population background. Differences in slope among clusters in Fig 2 further demonstrate contrasting responses to environmental and management conditions, highlighting variation in adaptability and yield stability.

Genotyping in Cluster 1 are dominantly drought tolerant and broadly adapted populations. Major contributors included pop4 (16.7%), pop1 (14.2%), pop2 (13.8%), pop5 (10.9) and pop12 (10.0%), together accounting for more than 40% of the cluster, along with 23.4% of genotypes of unknown background (Table 9). This composition explains the relatively stable but generally low mean yield across environments (Fig 1) based on the heatmap. It also explains the comparatively better performance under Kiboko_DT and the limited yield response under favorable or Kakamega conditions (Fig 2).

Cluster 2 contained lines derived from temperate introgression populations. The main contributors pop13 (15.3%), pop16 (13.9%), pop3 (13.6%), and pop11 (9.9%) represented nearly half of the cluster (Table 9). These elite backgrounds

**Table 9. Number and percentage of genotypes assigned to each genetic cluster based on GY, derived from doubled haploid (DH) lines originating from base populations with contrasting breeding characteristics.**

| Code | Pedigree | HG | Characteristic | Number of Genotypes per cluster and percentage | | | | | |
|------|----------|----|----------------|----------|---|----------|---|----------|---|
| | | | | Cluster1 | % | Cluster2 | % | Cluster3 | % |
| pop1 | CKDHL0500/CML202 | B | Drought tolerant by high yield | 34 | 14.2 | 4 | 1.4 | 11 | 3.4 |
| pop 2 | CKDHL0089/CML494 | B | High yield potential by MLN tolerant | 33 | 13.8 | 16 | 5.4 | 1 | 0.3 |
| pop 3 | CKDHL0089/CML543 | B | High yield potential by High yield potential | 6 | 2.5 | 40 | 13.6 | 2 | 0.6 |
| pop 4 | CKDHL0089/CKDHL120918 | B | High yield potential by MLN tolerant | 40 | 16.7 | 6 | 2.0 | 3 | 0.9 |
| pop 5 | CKLTI0139/CKDHL120918 | A | Temperate introgression by MLN-tolerant | 26 | 10.9 | 20 | 6.8 | 3 | 0.9 |
| pop 6 | CKDHL0228/CKLTI0133 | A | Elite line by temperate introgression | 10 | 4.2 | 18 | 6.1 | 22 | 6.7 |
| pop7 | CKLTI0133/CKDHL120312 | A | Temperate introgression by MLN-tolerant | 7 | 2.9 | 8 | 2.7 | 33 | 10.1 |
| pop 8 | CKLTI0227/CML543 | B | Temperate introgression by high-yield potential | 7 | 2.9 | 4 | 1.4 | 37 | 11.3 |
| pop 9 | CML395/CKLTI0052 | B | Drought tolerant by Temperate introgression | 18 | 7.5 | 6 | 2.0 | 25 | 7.6 |
| pop 10 | CKLTI0272/CKDHL0228 | A | Temperate introgression by elite line | 4 | 1.7 | 17 | 5.8 | 28 | 8.5 |
| pop 11 | CKLTI0344/CKLTI0147 | A | Temperate introgression by Temperate introgression | – | – | 29 | 9.9 | 21 | 6.4 |
| pop 12 | CML395/CKLMARSI0002 | B | Drought tolerant by drought tolerant | 24 | 10.0 | 1 | 0.3 | 23 | 7.0 |
| pop 13 | CKLTI0200/CML442 | A | Temperate introgression by drought-tolerant | 2 | 0.8 | 45 | 15.3 | – | – |
| pop 14 | CKLTI0200/CKLTI0036 | A | Temperate introgression by Temperate introgression | 5 | 2.1 | 7 | 2.4 | 41 | 12.5 |
| pop 15 | CKLTI0272/CML442 | A | Temperate introgressed line by drought-tolerant | 10 | 4.2 | 10 | 3.4 | 30 | 9.1 |
| pop 16 | CKLTI0368/CKDHL0228 | A | Temperate introgressed line by high-yielding line | 3 | 1.3 | 41 | 13.9 | 3 | 0.9 |
| pop 17 | DTMAMARSKEN5Bulk | A | Population developed through RCGS | 1 | 0.4 | 1 | 0.3 | – | – |
| pop 18 | DTMA MARS KEN6 Bulk | A | Population developed through RCGS | – | – | 2 | 0.7 | 2 | 0.6 |
| pop 19 | DTMA MARS KEN7 Bulk | A | Drought tolerant by drought tolerant | 1 | 0.4 | 10 | 3.4 | 33 | 10.1 |
| Mixed | unknown | | unknown | 4 | 1.7 | 8 | 2.7 | 9 | 2.7 |
| | | | Checks | 4 | 1.7 | 1 | 0.3 | 1 | 0.3 |
| **Total** | | | | **239** | | **294** | | **328** | |

HG = Heterotic Group; letters A and B represent Heterotic Group A and Heterotic Group B, respectively. 'Pop' followed by a number indicates the population number.

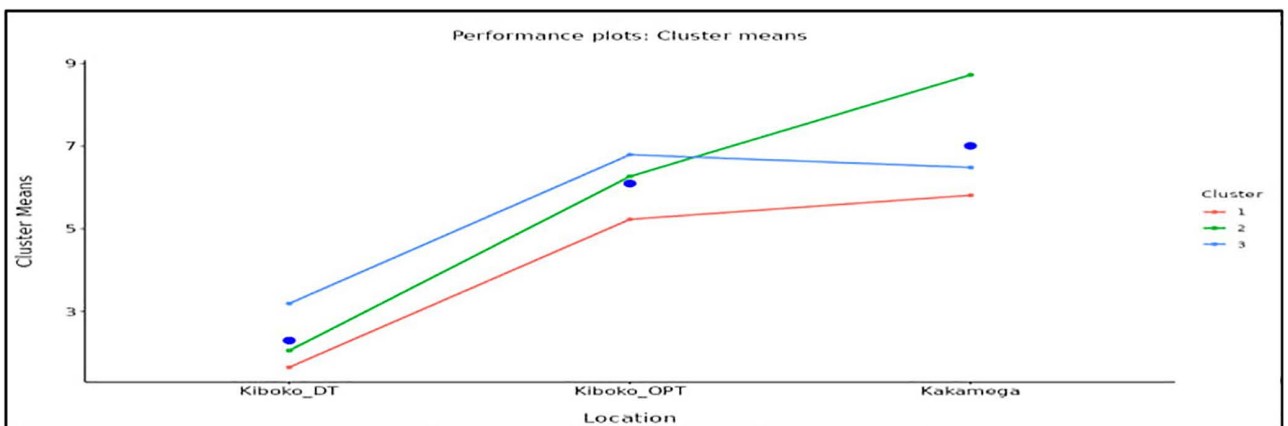

**Fig 2. Mean grain yield performance of the three genotype clusters across test environments.** The plot shows cluster mean yields under drought stress at Kiboko (Kiboko_DT), optimal conditions at Kiboko (Kiboko_OPT), and optimal conditions at Kakamega. Differences in slope among clusters indicate contrasting yield responses, stability, and levels of environmental adaptation under stress and favorable conditions.

explain the strong yield increase from drought to optimal conditions and the highest mean yield at Kakamega. The small contribution of the drought-tolerant populations suggests a limited representation or expression of drought-tolerance traits in their genetic background, as evidenced by their low yield performance under Kiboko_DT environment.

Cluster 3 was the largest, with 328 genotypes, and included populations combining temperate introgression, stress tolerance, and Molecular marker-based population improvement of elite lies. Pop14 (12.5%), pop8 (11.3%), pop7 (10.1%), pop15 (9.1%), and pop19 (10.1%) together contributed about 50% of the cluster, with additional representation from recurrent cycle genomic selection (RCGS) derived populations. This balanced composition explains the consistently strong performance across all environments. It also explains the improved drought tolerance relative to Cluster 2 and the modest yield reduction under optimal conditions.

Pairwise comparisons of the top 15% (128) selected genotypes across environments, based on BLUE mean values (Table 10), revealed low agreement among locations, confirming strong genotype × environment interaction. The highest overlap occurred between Kakamega_OPT and Kiboko_OPT, with 27 common genotypes (21.1%). Kakamega_OPT and Kiboko_DT shared 15 genotypes (11.7%). Only seven genotypes (5.5%) were common across all environments further emphasizes the strong genotype × environment interaction and, suggests limited stability of genotypic performance across diverse testing conditions.

Overall, yield-based clustering clearly differentiated genotypes by environmental adaptation. Cluster 1 showed low yield stability with better performance under drought. Cluster 2 displayed specific adaptation to favorable conditions at Kakamega. Cluster 3 showed strong and consistent performance across both optimal and drought environments at Kiboko. The close alignment among population composition, cluster mean yield, and selection overlap confirms that population formation strategy strongly influences genotype response, highlighting the complementary value of these clusters for breeding stress-tolerant, high-yielding, and broadly adapted maize.

### 3.6  Cluster-wise summary of top-performing genotypes

Cluster 1 was ranked using the mean yield across the three locations. Cluster 2 was ranked based on performance at Kakamega. Cluster 3 was ranked based on performance under drought stress at Kiboko_DT. This summary is based on the top five genotypes selected from each cluster according to yield performance across environments (Table 11).

In Cluster 2, evaluation under Kakamega conditions identified G135, G233, G849, G780, and G613 as the top-performing genotypes. These genotypes showed high grain yield and mainly originated from high-yielding, temperate-introgressed, and drought-tolerant populations. This indicates good adaptation to high-potential environments.

In Cluster 3, selection under Kiboko_DT conditions identified G576, G500, G707, G643, and G533 as the best-performing genotypes. These genotypes were largely derived from temperate-introgressed and elite-line backgrounds. Their performance indicates improved drought adaptation and yield stability under stress-prone conditions.

Overall, the top five genotypes from each cluster showed clear and distinct adaptation patterns. Cluster 2 was better suited to Kakamega environments. Cluster 3 showed stronger adaptation to Kiboko conditions. These genotypes provide valuable parental material for targeted breeding strategies.

### 3.7  Population-wise comparison based on top performing genotypes

The plot shows clear differences in grain yield and relative yield advantage among breeding populations under optimal and drought-stressed environments. Several populations derived from temperate introgressions and drought-tolerant × high-yielding crosses consistently outperformed the commercial checks. In some cases, yield advantages exceeded 30–53% (Fig 3). This indicates substantial genetic gain from targeted parental combinations. In contrast, populations with lower mean grain yield and little yield advantage show limited adaptation and narrower genetic potential across environments.

Table 10. The pairwise comparison of genotypes common among the top 15% (128 genotypes) selections from 855 total new hybrids across environments based on BLUE mean grain yield (ton/ha). The table lists genotype codes and their grain yield performance (t ha$^{-1}$) under optimal and drought stress condition. Genotypes are grouped according to their occurrence across all three locations. The number and percentage of common genotypes indicate the level of selection overlap between environments.

| Genotypes in common | Genotype Code | Kakamega_OPT | Kiboko_OPT | Kiboko_DT | # Genotype | % |
|---|---|---|---|---|---|---|
| Across three locations | G496 | 9.34 | 7.72 | 4.07 | 7 | 5.5 |
| | G508 | 9.92 | 7.99 | 4.45 | | |
| | G532 | 9.09 | 8.10 | 4.66 | | |
| | G571 | 8.97 | 8.04 | 4.22 | | |
| | G581 | 10.35 | 7.98 | 4.25 | | |
| | G583 | 10.32 | 8.20 | 4.10 | | |
| | G585 | 10.84 | 7.85 | 4.07 | | |
| | G543 | 8.92 | – | 3.71 | 15 = (7 + 8) | 11.7 |
| | G547 | 8.89 | – | 3.70 | | |
| Kakamega_OPT and Kiboko_DT | G557 | 10.49 | – | 3.90 | | |
| | G569 | 10.50 | – | 4.13 | | |
| | G582 | 10.39 | – | 4.28 | | |
| | G587 | 9.03 | – | 3.64 | | |
| | G781 | 9.24 | – | 3.62 | | |
| | G788 | 9.51 | – | 3.74 | | |
| | G113 | 10.87 | 8.08 | – | 27 = (20 + 7) | 21.1 |
| | G116 | 10.34 | 8.70 | – | | |
| | G120 | 9.69 | 7.88 | – | | |
| | G121 | 9.21 | 7.92 | – | | |
| Kakamega_OPT and Kiboko_OPT | G123 | 10.44 | 8.01 | – | | |
| | G130 | 10.73 | 7.53 | – | | |
| | G135 | 12.67 | 8.00 | – | | |
| | G143 | 9.05 | 7.53 | – | | |
| | G144 | 9.56 | 9.01 | – | | |
| | G150 | 10.07 | 8.39 | – | | |
| | G285 | 10.19 | 8.07 | – | | |
| | G551 | 10.64 | 7.76 | – | | |
| | G570 | 9.07 | 7.73 | – | | |
| | G578 | 9.53 | 7.81 | – | | |
| | G593 | 10.48 | 7.88 | – | | |
| | G596 | 10.36 | 8.57 | – | | |
| | G605 | 10.23 | 7.51 | – | | |
| | G619 | 10.29 | 7.50 | – | | |
| | G634 | 9.24 | 7.83 | – | | |
| | G849 | 11.90 | 7.87 | – | | |

Where, OPT= Well water or Optimum, DT= Drought.

## 3.8 Multivariate consideration for clustering genotypes across environments

K-means clustering based on multi-trait data was evaluated using the elbow method (Fig 4). The within-cluster sum of squares declined sharply as the number of clusters increased from one to three, indicating a strong improvement in cluster compactness. Beyond three to four clusters, the reduction in within-cluster variation became small and gradual.

**Table 11. Mean grain yield performance of the top five genotypes selected from each cluster across three environments (Kakamega, Kiboko_OPT, and Kiboko_DT), together with their genotype codes, pedigree of original populations, population characteristics, and population identifiers.**

| Cluster | Genotype | Code | Kakamega | Kiboko_OPT | Kiboko_DT | Mean | Pedigree of base population | Characteristic of population | population# |
|---|---|---|---|---|---|---|---|---|---|
| 1 | CKDHL1706211 | G251 | 6.87 | 5.88 | 2.2 | **4.98** | CKDHL0228/ CKLTI0133 | Elite line by temperate introgression | 6 |
| 1 | CKDHL1719170 | G757 | 7.87 | 4.83 | 2.25 | **4.98** | CKLTI0368/ CKDHL0228 | Temperate introgressed line by high-yielding line | 16 |
| 1 | CKDHL1702568 | G156 | 7.03 | 6.02 | 1.8 | **4.95** | CKDHL0089/ CKDHL120918 | High yield potential by MLN tolerant | 4 |
| 1 | CKDHL1710527 | G416 | 6.44 | 6.63 | 1.64 | **4.90** | CML395/ CKLTI0052 | Drought tolerant by Temperate introgression | 9 |
| 1 | CKDHL1700018 | G8 | 7.03 | 5.8 | 1.85 | **4.89** | CKDHL0500/ CML202 | Drought tolerant by high yield | 1 |
| 2 | CKDHL1702100 | G135 | **12.67** | 8.00 | 1.32 | 7.33 | CKDHL0089/ CML543 | High yield potential by High yield potential | 3 |
| 2 | CKDHL1705916 | G233 | **11.96** | 5.37 | 1.06 | 6.13 | CKLTI0139/ CKDHL120918 | Temperate introgression by MLN-tolerant | 5 |
| 2 | CKLTI0368 | G849 | **11.90** | 7.87 | 2.11 | 7.29 | Mixed | unknown | Mixed |
| 2 | CKDHL1719346 | G780 | **11.65** | 4.78 | 3.22 | 6.55 | CKLTI0368/ CKDHL0228 | Temperate introgressed line by high-yielding line | 16 |
| 2 | CKDHL1717115 | G613 | **11.59** | 6.66 | 1.08 | 6.44 | CKLTI0200/ CML442 | Temperate introgression by drought-tolerant | 13 |
| 3 | CKDHL1715839 | G576 | 6.49 | 8.06 | **6.13** | 6.89 | CKLTI0344/ CKLTI0147 | Temperate introgression by Temperate introgression | 11 |
| 3 | CKDHL1715313 | G500 | 6.79 | 6.78 | **5.80** | 6.46 | CKLTI0272/ CKDHL0228 | Temperate introgression by elite line | 10 |
| 3 | CKDHL1718483 | G707 | 8.65 | 6.63 | **5.16** | 6.81 | CKLTI0272/ CML442 | Temperate introgressed line by drought-tolerant | 15 |
| 3 | CKDHL1717281 | G643 | 6.62 | 6.75 | **5.14** | 6.17 | CKLTI0200/ CKLTI0036 | Temperate introgression by Temperate introgression | 14 |
| 3 | CKDHL1715563 | G533 | 7.74 | 7.31 | **4.99** | 6.68 | CKLTI0272/ CKDHL0228 | Temperate introgression by elite line | 10 |

The curve flattened after four clusters, showing diminishing returns from adding more clusters. The clear elbow at K = 4 suggests an optimal balance between explained variability and model simplicity. Increasing cluster numbers beyond this point did not meaningfully improve within-cluster homogeneity. Therefore, the data are best represented by four distinct clusters, which were used for downstream interpretation and biological inference.

Multivariate diversity analysis was performed using combined data across locations and management conditions (optimum and drought). Cluster and pattern analyses grouped the 861 maize genotypes into four distinct clusters: C1 (267 genotypes), C2 (125), C3 (245), and C4 (224) (Fig 5). The dendrogram revealed strong associations among key traits. Grain yield (GY) was closely associated with ears per plant (EPP). Silking date (SD) showed a strong relationship with AD, indicating shared genetic control and coordinated trait expression (Fig 5).

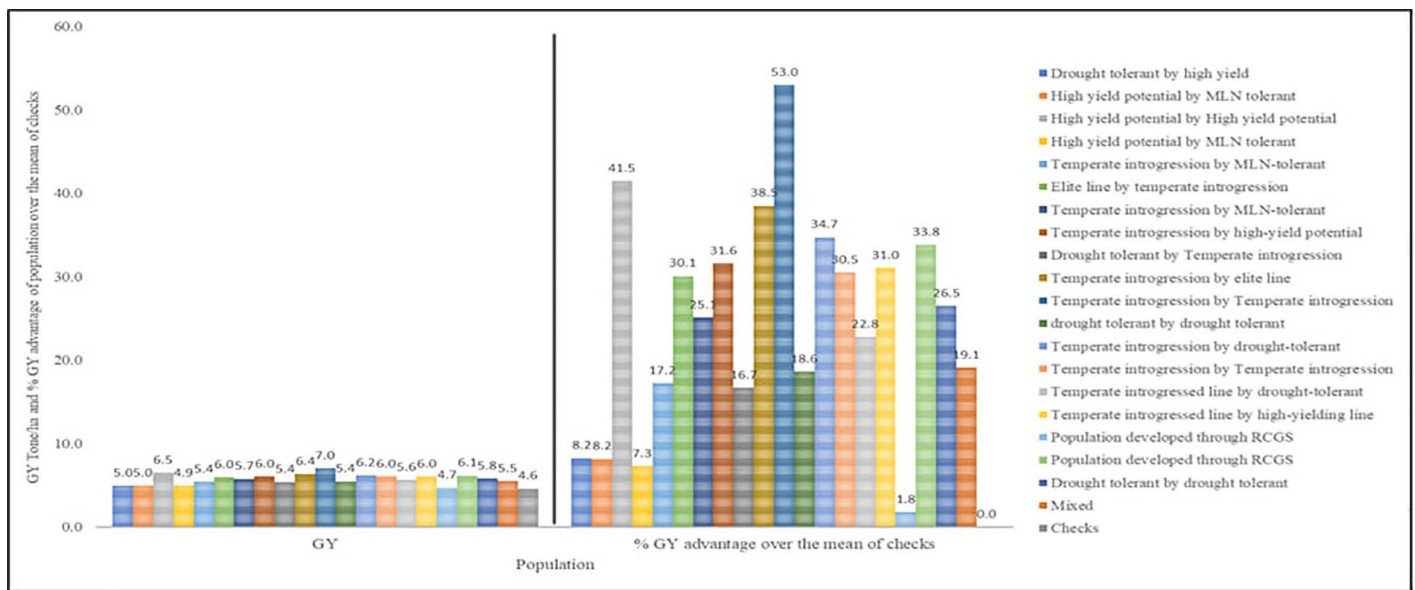

**Fig 3. Mean grain yield (GY) and percentage yield advantage relative to the mean of check varieties of the top 20 genotypes from each breeding population evaluated across three environments (Kakamega_OPT, Kiboko_OPT, and Kiboko_DT).**

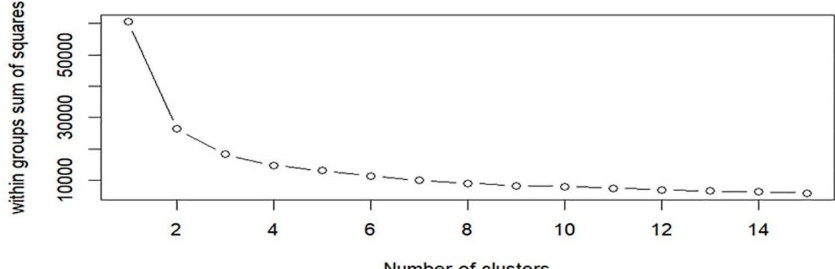

**Fig 4. Elbow plot for determining the optimal number of clusters using the K-means algorithm.** The figure shows the relationship between the number of clusters (K) and the within-cluster sum of squares. A steep reduction in within-cluster variation is observed as K increases from 1 to 3, indicating improved cluster compactness. The curve begins to level off at K=4, forming a clear elbow point, beyond which additional clusters result in only marginal reductions in within-cluster sum of squares. This pattern indicates that four clusters adequately capture the underlying structure of the data and were therefore selected for subsequent clustering and interpretation.

### 3.9 Integration of PCA clustering and multi-trait heatmap

The combined interpretation of the PCA clustering table (Table 12) and the multi-trait heatmap (Fig 5) provides a clear understanding of how source population characteristics shaped the genetic structure of the germplasm. Table 12 quantified the number and percentage of genotypes contributed by each population to the four clusters. The heatmap visually displayed the trait profiles underlying this classification. Together, these results provide numerical and graphical validation of the clustering pattern.

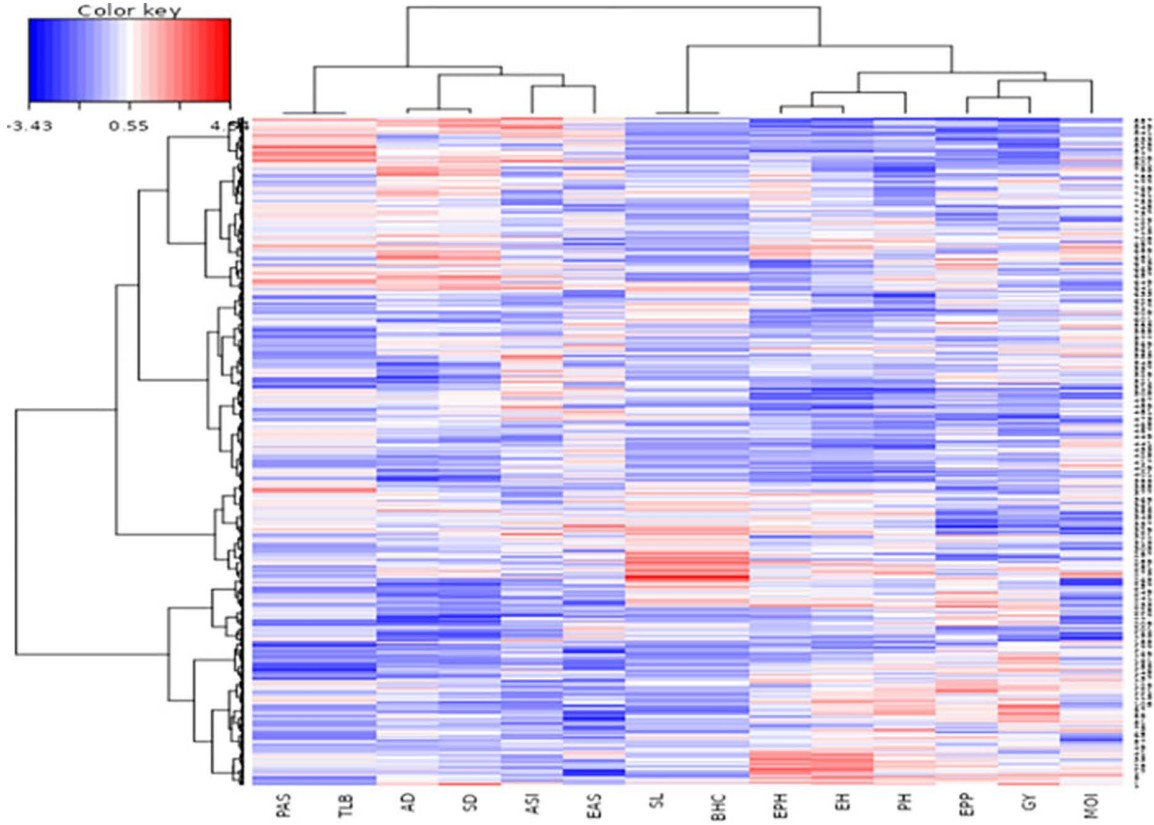

**Fig 5. Hierarchical clustering (dendrogram) of maize genotypes and agronomic traits based on multi-trait phenotypic data.** Rows correspond to individual genotypes, while columns represent measured traits (where PAS = Plant Aspect, EAS = Ear Aspect, ASI = Anthesis Silking Interval, SD Silking Date, AD Anthesis Date, MOI = Grain Moisture Content, GY = Grain Yield, EPP = Ear Per Plant, PH = Plant Height, EH = Ear Height, EPH = Ear Position Height, SL = Stalk Lodging and BHC = Bad Husk Cover, TLB = Turcicum Leaf Blight). The dendrograms illustrate similarity relationships among genotypes and traits derived from standardized phenotypic values across environments. Color intensity reflects relative trait expression, where red indicates above-average values, blue indicates below-average values, and white represents values close to the overall mean, as shown in the color key. This heatmap reveals trait associations, genotype differentiation, and contrasting multi-trait performance patterns, supporting the identification of favorable trait combinations for selection.

The PCA separated genotypes mainly along PC1 and PC2. PC1 was driven by yield and plant architecture traits. PC2 captured phenology and stress-related traits. This structure was reflected in the heatmap, where hierarchical clustering divided genotypes into four major blocks with contrasting color patterns. These blocks closely corresponded to Clusters C1- C4, confirming that genotypes with similar breeding histories showed similar multivariate trait performance.

Cluster 1 contained 267 genotypes and was dominated by populations with temperate introgression and elite tropical backgrounds, including pop6, pop7, pop10 and pop11. In the heatmap, this cluster showed strong expression of GY, EPP, PH, and EH. This consistency indicates that Cluster 1 represents broadly adapted, high-yielding germplasm suitable for mainstream hybrid development.

Cluster 2 comprised 125 genotypes and was mainly represented by Pop2 and Pop3, each contributing 26 genotypes (20.8%). These populations were developed for high yield and MLN tolerance under optimal conditions. The heatmap supported this pattern by showing lower expression of yield-related traits and weaker responses for

**Table 12. Number and percentage of genotypes assigned to each genetic cluster based on multi trait consideration, derived from doubled haploid (DH) lines originating from base populations with contrasting breeding characteristics.**

| Code | Pedigree | HG | Characteristic | Cluster1 | % | Cluster2 | % | Cluster3 | % | Cluster4 | % |
|------|----------|----|----------------|----------|---|----------|---|----------|---|----------|---|
| | | | | **Number of Genotypes per cluster and percentage** | | | | | | | |
| pop1 | CKDHL0500/CML202 | B | Drought tolerant by high yield | – | – | 1 | 0.8 | 13 | 5.3 | 35 | 15.6 |
| pop 2 | CKDHL0089/CML494 | B | High yield potential by MLN tolerant | 6 | 2.2 | 26 | 20.8 | 13 | 5.3 | 5 | 2.2 |
| pop 3 | CKDHL0089/CML543 | B | High yield potential by High yield potential | 9 | 3.4 | 26 | 20.8 | 12 | 4.9 | 1 | 0.4 |
| pop 4 | CKDHL0089/ CKDHL120918 | B | High yield potential by MLN tolerant | 1 | 0.4 | 15 | 12.0 | 11 | 4.5 | 22 | 9.8 |
| pop 5 | CKLTI0139/ CKDHL120918 | A | Temperate introgression by MLN-tolerant | 5 | 1.9 | 4 | 3.2 | 26 | 10.6 | 14 | 6.3 |
| pop 6 | CKDHL0228/ CKLTI0133 | A | Elite line by temperate introgression | 38 | 14.2 | 4 | 3.2 | 2 | 0.8 | 6 | 2.7 |
| pop7 | CKLTI0133/ CKDHL120312 | A | Temperate introgression by MLN-tolerant | 43 | 16.1 | 4 | 3.2 | 1 | 0.4 | – | – |
| pop 8 | CKLTI0227/CML543 | B | Temperate introgression by high-yield potential | 17 | 6.4 | 2 | 1.6 | 25 | 10.2 | 4 | 1.8 |
| pop 9 | CML395/CKLTI0052 | B | Drought tolerant by Temperate introgression | 8 | 3.0 | 16 | 12.8 | 18 | 7.3 | 7 | 3.1 |
| pop 10 | CKLTI0272/ CKDHL0228 | A | Temperate introgression by elite line | 29 | 10.9 | 4 | 3.2 | 10 | 4.1 | 6 | 2.7 |
| pop 11 | CKLTI0344/ CKLTI0147 | A | Temperate introgression by Temperate introgression | 47 | 17.6 | – | – | 2 | 0.8 | 1 | 0.4 |
| pop 12 | CML395/ CKLMARSI0002 | B | Drought tolerant by drought tolerant | – | – | 1 | 0.8 | 5 | 2.0 | 42 | 18.8 |
| pop 13 | CKLTI0200/CML442 | A | Temperate introgression by drought-tolerant | 9 | 3.4 | 5 | 4.0 | 17 | 6.9 | 16 | 7.1 |
| pop 14 | CKLTI0200/ CKLTI0036 | A | Temperate introgression by Temperate introgression | 10 | 3.7 | 10 | 8.0 | 19 | 7.8 | 14 | 6.3 |
| pop 15 | CKLTI0272/CML442 | A | Temperate introgressed line by drought-tolerant | 6 | 2.2 | – | – | 38 | 15.5 | 6 | 2.7 |
| pop 16 | CKLTI0368/ CKDHL0228 | A | Temperate introgressed line by high-yielding line | 25 | 9.4 | 3 | 2.4 | 19 | 7.8 | – | – |
| pop 17 | DTMAMARSKEN5Bulk | A | Population developed through RCGS | – | – | – | – | – | – | 2 | 0.9 |
| pop 18 | DTMA MARS KEN6 Bulk | A | Population developed through RCGS | – | – | – | – | – | – | 4 | 1.8 |
| pop 19 | DTMA MARS KEN7 Bulk | A | Drought tolerant by drought tolerant | 1 | 0.4 | 1 | 0.8 | 8 | 3.3 | 34 | 15.2 |
| Mixed | unknown | | unknown | 10 | 3.7 | 2 | 1.6 | 5 | 2.0 | 4 | 1.8 |
| | | | Checks | 3 | 1.1 | 1 | 0.8 | 1 | 0.4 | 1 | 0.4 |
| **Total** | | | | **267** | | **125** | | **245** | | **224** | |

HG = Heterotic Group; letters A and B represeant Heterotic Group A and Heterotic Group B, respectively. 'Pop' followed by a number indicates the population number.

stress-related traits, such as ASI and SD, together with higher values for stem lodging (SL) and bare husk cover (BHC).

Cluster 3 included 245 genotypes with major contributions from pop8, pop5, pop13, pop14, and pop15, with pop15 contributing the most. The heatmap showed reduced yield and plant architecture traits but stronger expression of stress-adaptive traits, including shorter ASI and earlier flowering and maturity. The presence of known drought-tolerant populations, such as pop1 and pop12, indicates that this cluster represents a diverse stress-adapted group for drought-prone environments.

Cluster 4 contained 224 genotypes and was enriched with drought-targeted particularly pop1, pop12, pop19 populations. In the heatmap, this cluster formed a distinct block with strong stress tolerance traits but reduced performance for optimum yield traits. This pattern indicates that Cluster 4 represents highly drought-resilient germplasm selected under low-moisture conditions and is valuable for breeding in marginal environments. Genotypes from pop17 and 18, each represented by very few genotypes (n = 2 and n = 4, respectively) clustered only under cluster 4, showing limited distribution across clusters (Table 12), which likely reflects their small sample size and reduced representation of within-population variability.

Finally, the broad distribution of the uncharacterized mixed population across all clusters in Table 12 was reflected by its scattered pattern in the heatmap. This highlights the high genetic diversity of the panel. It also demonstrates the effectiveness of combining PCA and heatmap analyses for organizing complex maize germplasm.

### 3.10 Cluster mean performance and variability

The clustering of genotypes revealed important patterns for breeding (Table 13; Fig 5). Clusters differed for yield-related traits, with some clusters showing higher performance than others. Variation among clusters for GY, PH, and EH confirms sufficient genetic diversity to support multi-trait selection. Traits with minimal differences, such as EAS, PAS, and TLB, showed low variability, indicating high uniformity within clusters.

**Table 13. Cluster-wise mean and standard deviation of grain yield, phenological, agronomic, and plant architectural traits of maize genotypes based on K-means clustering using standardized multi-trait phenotypic data.**

| Mean | | | | | | | | | | | | | | |
|---|---|---|---|---|---|---|---|---|---|---|---|---|---|---|
| Cluster | GY | AD | EPH | EPP | SL | BHC | EAS | PAS | TLB | MOI | PH | EH | SD | ASI |
| 1 | 5.40 | 65.67 | 0.52 | 0.93 | 11.18 | 11.18 | 2.83 | 2.69 | 2.69 | 20.01 | 239.32 | 125.83 | 66.60 | 1.02 |
| 2 | 5.17 | 66.62 | 0.53 | 0.90 | 13.15 | 13.15 | 2.97 | 2.72 | 2.72 | 20.00 | 238.53 | 125.27 | 67.96 | 1.29 |
| 3 | 5.18 | 65.88 | 0.51 | 0.91 | 11.15 | 11.15 | 2.96 | 2.70 | 2.70 | 20.09 | 233.04 | 117.35 | 67.25 | 1.30 |
| 4 | 5.15 | 67.44 | 0.52 | 0.91 | 10.91 | 10.91 | 2.96 | 2.74 | 2.74 | 20.04 | 234.77 | 120.47 | 68.90 | 1.35 |
| | Standard deviation | | | | | | | | | | | | | |
| | GY | AD | EPH | EPP | SL | BHC | EAS | PAS | TLB | MOI | PH | EH | SD | ASI |
| 1 | 0.22 | 1.14 | 0.02 | 0.02 | 1.14 | 1.14 | 0.15 | 0.04 | 0.04 | 0.26 | 4.11 | 6.03 | 1.23 | 0.28 |
| 2 | 0.20 | 0.83 | 0.02 | 0.02 | 1.17 | 1.17 | 0.12 | 0.04 | 0.04 | 0.25 | 3.72 | 4.37 | 0.89 | 0.34 |
| 3 | 0.18 | 1.04 | 0.02 | 0.02 | 0.93 | 0.93 | 0.12 | 0.04 | 0.04 | 0.22 | 4.30 | 4.61 | 1.09 | 0.37 |
| 4 | 0.22 | 0.96 | 0.02 | 0.02 | 0.94 | 0.94 | 0.12 | 0.04 | 0.04 | 0.24 | 5.15 | 5.35 | 1.09 | 0.40 |

Note: The mean values describe the average performance of each cluster for the evaluated traits, while the corresponding standard deviations indicate the degree of within-cluster variability. Where GY = Grain yield, AD = Anthesis date, SD = Silking date, ASI = Anthesis Silking interval, PH = Plant Height, EH = Ear Height, EPH = Ear Position Height, EPP = Ear Per Plant, SL = Stalk Lodging, BHC = Bad Husk Cover, EAS = Ear Aspect, PAS = Plant Aspect, MOI = Grain Moisture, TLB = Turcicum Leaf Blight.

Days to anthesis differed slightly among clusters. Some clusters flowered earlier, while others were later. The ASI increased across clusters, with the shortest interval observed in the earliest-flowering cluster, which is desirable for reproductive synchrony and stress tolerance. Low variability in phenological traits suggests consistency within clusters.

Plant height and EH contributed noticeably to cluster differentiation. Some clusters had taller plants with higher ear placement, while others were shorter. Structural traits, including stalk lodging and BHC, showed minor differences, with certain clusters exhibiting slightly better structural attributes. Yield-related and quality traits were generally uniform across clusters.

The heatmap (Fig 5) visualizes these patterns, showing clear clustering of traits and genotypes. Higher values for key traits appear prominently, while lower values are distinguishable, highlighting trait differentiation among clusters.

Overall, clusters differed in agronomic performance. Some combined high yield, early flowering, and short ASI, making them desirable for breeding. Others showed balanced performance or distinct structural traits. Low within-cluster variability confirms effective clustering, and the observed inter-cluster differences provide opportunities for parent selection and hybrid development in maize breeding programs.

### 3.11 Biplot pattern analysis (PCA) in multivariate

In the PCA summary, to achieve 100% cumulative variance, 11 principal components were needed. The first three components accounted for 61.6% of the variation, with their standard deviations/eigenvalues greater than one. In contrast, the remaining eight components had standard deviation values below one, indicating that they were less significant and explain less variation. Specifically, the first two components together accounted for 43.7% of the total variation, with PC1 explaining 25.4% and PC2 explaining 18.3%. The biplot from the multivariate analysis revealed four distinct genotype clustering patterns (Fig 6).

The length of trait vectors in the PCA biplot reflects their contribution to the total variation explained by the first two principal components. Traits with longer vectors in Fig 6, particularly GY and key plant architectural traits, contributed most to genotype discrimination and were the primary drivers of phenotypic diversity. Specifically, GY exhibited a longer vector than EPP along both PC1 and PC2, indicating that GY has stronger influence on the variation explained by the first two principal components. In contrast, EPP and other traits with relatively shorter vectors contributed less to genotype separation, reflecting lower variability or weaker association with PC1 and PC2. Notably, the MOI displayed the shortest vector length, indicating that it contributed the least to the variation captured by PC1 and PC2.

### 3.12 Pearson correlation analysis

The Pearson correlation matrix (Fig 7) illustrates the relationships among GY and the evaluated agronomic traits. Grain yield exhibited moderate positive correlations with PH; $r \approx 0.39$, EH; $r \approx 0.39$, EPP; $r \approx 0.52$, and MOI; $r \approx 0.22$, indicating that increases in these traits tend to be associated with higher grain yield. In contrast, GY showed negative correlations with AD; $r \approx -0.09$, SD; $r \approx -0.21$, ASI; $r \approx -0.25$, EAS; $r \approx -0.45$, and PAS; $r \approx -0.46$, suggesting that delayed flowering, wider ASI, and poorer ear and plant appearance are associated with reduced yield.

Anthesis date was strongly positively correlated with SD ($r \approx 0.89$) and moderately correlated with ASI, EH, EPH, EAS, PAS, and MOI, while showing a weak negative association with EPP. ASI displayed negative correlations with PH, EH, EPH, EPP, EAS, and PAS, indicating that a longer ASI tends to coincide with reduced plant vigor and yield-related traits. Plant height was positively correlated with EH, EPH, and EPP, whereas negative associations were observed with EAS and PAS. Ear height similarly showed positive correlations with EPH, EPP, and MOI, and negative correlations with EAS and PAS. In general, the correlation pattern highlights that traits such as ears per plant, ear placement traits, plant height, and moisture index are favorably associated with grain yield, while delayed flowering, extended ASI, and poor ear and plant aspects show unfavorable relationships with yield. These associations underscore the importance of jointly considering yield-enhancing and yield-limiting traits in selection decisions.

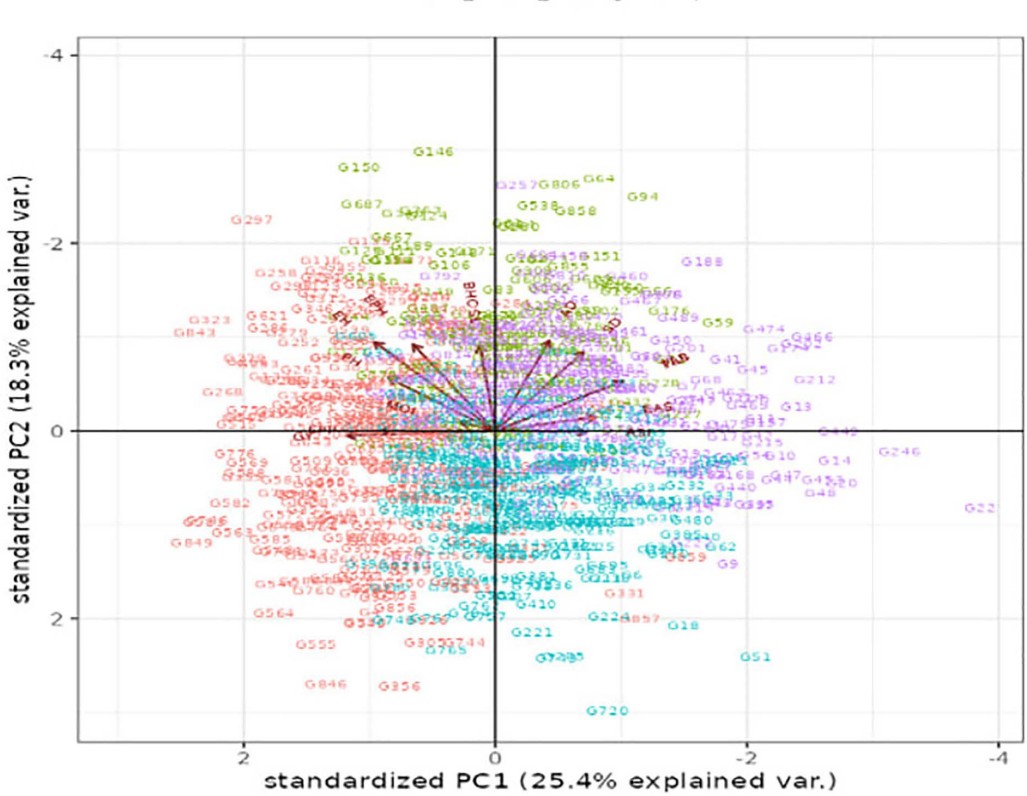

**Fig 6. Principal component analysis (PCA) biplot showing the distribution of genotypes and trait contributions based on standardized multi-trait phenotypic data.** The first two principal components explained 25.4% (PC1) and 18.3% (PC2) of the total phenotypic variation. Each point represents a genotype, colored according to its K-means cluster membership (Clusters 1–4), while labels denote individual genotypes. The arrows indicate trait loadings and their direction and length reflect the magnitude and contribution of each trait to the observed variation. Genotypes located close to the origin exhibit average performance, whereas those farther from the center show stronger associations with specific traits. The four quadrants highlight distinct patterns of trait association and genotype grouping, demonstrating clear multivariate differentiation among clusters and confirming the effectiveness of clustering based on combined agronomic and phenological traits. Where GY = Grain Yield, AD = Anthesis date, SD = Silking date, ASI = Anthesis Silking interval, PH = Plant Height, EH = Ear Height, EPH = Ear Position Height, EPP = Ear Per Plant, BHC = Bad Husk Cover, SL = Stalk Lodging, EAS = Ear Aspect, MOI = Grain Moisture Content.

## 4 Discussion

### 4.1 Phenotypic variation and heritability

Phenotypic analysis revealed substantial variation among maize genotypes under both optimum and moisture-stressed conditions. Significant differences in grain yield across locations indicated inconsistent genotype performance. Similar results have been reported under drought conditions by Poudel et al. [31] and Dube et al. [32]. Narrow-sense heritability for GY, AD, PH, and EH was consistently higher under drought conditions. This indicates stronger additive genetic control in stress environments. The higher heritability observed under moisture stress at Kiboko suggests that genetic variance was more effectively expressed relative to environmental variance. As a result, genetic differences among genotypes became more detectable, improving selection efficiency.

Similar findings have been reported in tropical maize, where moderate-to-high heritability under managed drought enabled effective selection for grain yield and stress tolerance [33–35]. These studies highlight that well-managed stress conditions can enhance the expression of additive genetic effects and accelerate genetic gain.

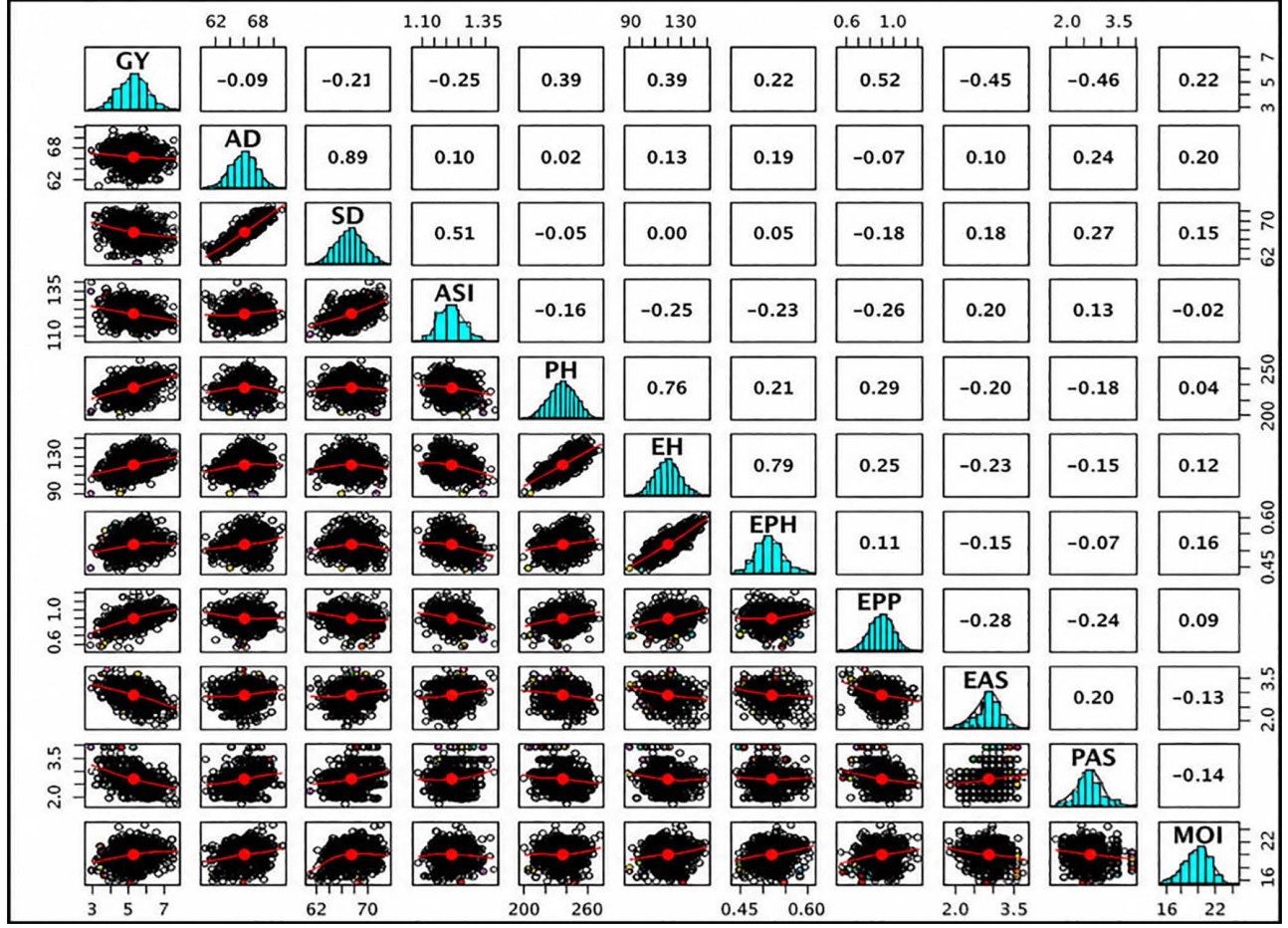

**Fig 7. Scatterplot matrix showing Pearson's correlation coefficients between GY and key agronomic traits of test crosses evaluated across three locations.** Diagonal panels display the frequency distributions of individual traits, lower panels show pairwise scatterplots with fitted trend lines, and upper panels present Pearson's correlation coefficients. Positive and negative values indicate the direction and strength of associations between traits, highlighting key relationships influencing grain yield performance across environments. Where GY = Grain Yield, AD = Anthesis date, SD = Silking date, ASI = Anthesis Silking interval, PH = Plant Height, EH = Ear Height, EPH = Ear Position Height, EPP = Ear Per Plant, EAS = Ear Aspect, PAS = Plant Aspect, MOI = Grain Moisture Content.

However, contrasting evidence also exists. Some studies report lower heritability under drought due to strong genotype × environment interaction and the complex genetic basis of stress tolerance [36–38]. Under severe or highly variable stress, environmental variance may dominate and reduce the expression of genetic potential. Similar trends were observed by [39] and Beyene et al. [40], where higher heritability was reported under optimal conditions due to more stable environments.

These differences indicate that heritability under drought is highly context-dependent. It is influenced by stress intensity, environmental uniformity, and population structure. In the present study, drought conditions at Kiboko were sufficiently discriminative to reveal additive genetic variation without masking phenotypic expression.

From a breeding perspective, these results emphasize the importance of selection under target stress environments. This approach improves the identification of genotypes with true drought tolerance. At the same time, multi-environment testing remains essential to ensure stable performance across diverse conditions. The CV for grain yield was highest

under the Kiboko drought-stressed condition (37.73%), indicating increased variability under water-limited environments. This is consistent with previous studies in maize, where the coefficient of variation (CV) for grain yield is typically higher under drought stress than under well-watered conditions and can range from approximately 25.6% to 48.08% across different testing locations under water-deficit environments reported by Ningning et al. [41]. Such high variability reflects differential genotypic responses to stress and the reduction in mean yield under drought conditions.

### 4.2 Genotype variance and environmental effects

Under optimum condition significant genotypic variance and moderate heritability estimates observed for grain yield and related traits under optimum conditions suggest the presence of exploitable additive genetic variation. This aligns with recent studies showing that substantial genetic variability for yield and agronomic traits exists among maize genotypes under well-watered environments [42]. Such variability offers opportunities for effective phenotypic selection and genetic gain. The significant genotype × location interaction effects indicate environmental influence on trait expression, consistent with evidence that both genotype and environment contribute to yield variation in maize breeding evaluations [43]. These results reaffirm that stable genetic differences under favorable conditions can guide breeders in selecting high-performing genotypes.

Under moisture stress, the presence of notable genotypic variance and moderate to high heritability for yield and related traits highlights the potential for genetic improvement even in challenging environments. Recent research has similarly reported that genetic variance and heritable differences can be detected for key traits under water deficit, supporting the feasibility of selection under drought [42]. Significant genotype × location interaction under stress further underscores how genotype responses vary with environmental conditions, a pattern consistently observed in maize evaluations across contrasting moisture regimes [44]. The notable residual variance for some traits suggests complex stress responses that may require careful evaluation across seasons to reliably select superior genotypes.

The residual variance for GY and ASI was higher under well-watered (optimum conditions), by 22 and 38%, respectively, as compared to under drought stress condition. The increase might be associated with the greater environmental heterogeneity in the combined optimum analysis. The optimum dataset included both Kakamega and Kiboko, which differ markedly in temperature, humidity, and evaporative demand. Such contrasts likely increased unexplained variability in ASI, a trait known to be highly sensitive to environmental fluctuations [27,45]. In contrast, the managed drought stress at Kiboko provided a more uniform environment, reducing environmental noise and resulting in lower residual variance. This suggests that ASI is more environmentally plastic under heterogeneous optimum conditions than under controlled stress.

### 4.3 Performance of DH hybrids and standard heterosis

The mean of top 10 hybrids showed strong grain yield performance compared with the mean of commercial checks, with average superiority of 161% under drought and 60.4% under optimum conditions. When compared with previous tropical doubled haploid (DH) studies, these gains appear exceptionally high. For instance, DH-derived maize hybrids evaluated in East Africa typically showed yield advantages ranging from about 23% under optimum conditions to approximately 43% under drought stress relative to commercial checks [46]. Similarly, other tropical hybrid improvement studies reported yield gains of 21–23% under optimum conditions and 51–52% under drought stress for top-performing entries [15]. Therefore, the 161% superiority observed under drought in the present study substantially exceeds the range commonly reported, while the 60.4% gain under optimum conditions is also notably higher than typical values. Recent studies also report that newly developed maize hybrids can achieve high yield and wide adaptability across different environments [47]. The variation in heterosis among crosses indicates the presence of genotype × environment interaction. This interaction is widely reported in maize breeding [48]. Molecular studies further show that heterotic expression can change with environmental conditions. Over dominant gene expression contributes to hybrid vigor in specific environments [49]. These findings highlight the importance of multi-environment testing to identify stable and high-performing hybrids.

Under optimum conditions, several hybrids showed positive standard heterosis for grain yield. This confirms the effectiveness of heterosis breeding in maize. The superiority of these hybrids reflects non-additive gene action and good parental complementarity [50,51]. Hybrids with high standard heterosis under favorable conditions are suitable for further multi-location evaluation.

Under drought conditions, high and significant standard heterosis was also observed in selected hybrids. This indicates that heterosis plays an important role in stress adaptation and yield improvement under moisture stress. Similar results have been reported in maize breeding programs targeting drought-prone environments [52]. Recent molecular evidence suggests that over dominant gene expression and stress-responsive pathways contribute to drought-related heterosis [49]. Therefore, the identified genotypes are valuable resources for drought-focused maize breeding programs.

### 4.4 Multivariate analysis and clustering

Principal component analysis captured most of the phenotypic variation and grouped genotypes into four distinct clusters, consistent with [53]. Multivariate clustering provided better resolution of diversity than single-trait analysis and helped to identify genotypes suitable for both high yield and stress tolerance. The clustering analysis, considering multiple agronomic and yield-related traits simultaneously, revealed genetically distinct groups with specific trait combinations. This pattern aligns with studies showing significant phenotypic and genetic divergence among maize genotypes for traits such as grain yield, anthesis-silking interval (ASI), plant height, and other yield components, where distinct clusters often reflect biologically meaningful diversity useful for breeding [32,36,54].

The ASI ranged narrowly from 1.02 to 1.35 days across all clusters (Table 13), indicating strong synchrony between pollen shed and silk emergence among the evaluated hybrids. This consistently short ASI is physiologically important and widely recognized as a critical driver of drought resilience in maize. Under water stress, silk emergence is typically delayed more than anthesis, resulting in an extended ASI, reduced pollination success, and poor kernel set. In contrast, genotypes that maintain a short ASI, as observed in this study, preserve reproductive synchrony and ensure effective fertilization even under moisture-limited conditions [27]. Recent studies further confirm that ASI is a key indicator of stress sensitivity, where increased ASI reflects disrupted reproductive development and consequent yield reduction under drought stress [45,55]. This reinforces ASI as a robust physiological marker linking reproductive failure to yield loss under water-limited environments. The consistently short ASI identified in this study expressed alongside other favorable agronomic and physiological traits that contribute to the overall adaptation. This suggests that drought resilience in these hybrids is not governed by a single trait, rather by a coordinated suite of traits that collectively enhance reproductive stability and yield performance.

### 4.5 Population performance and breeding implications

Our clustering and PCA analyses revealed clear differences in population performance across environments, emphasizing the influence of population origin and breeding history on yield and stress adaptation. Based on GY grouping, Cluster 1 dominated by drought-oriented and broadly adapted populations, exhibited stable but moderate yields, performing relatively better under stress, whereas Cluster 2, enriched with temperate introgression and elite yield-focused populations, showed strong performance under optimal conditions but limited drought tolerance. Cluster 3 combined populations with stress tolerance and advanced selection strategies, maintaining high performance across all environments, indicating broader adaptability. These patterns are consistent with previous studies showing that population-specific breeding strategies shape genotype × environment responses, with temperate introgression and elite tropical populations often driving higher yields under favorable conditions, while stress-adapted populations maintain performance under drought [56,57].

The low overlap of top-performing genotypes across environments highlights substantial genotype × environment interaction, reinforcing the need for targeted selection based on population-specific performance rather than single-genotype evaluation. This aligns with prior findings that multilocation evaluation and multivariate analysis provide better resolution

of both broadly and specifically adapted germplasm, enabling the identification of populations with high yield potential or stress resilience [58,59]. From a breeding perspective, these results underscore the value of combining population performance data, clustering, and PCA to guide parental selection and hybrid development. Specifically, the complementary strengths of clusters broad adaptability, stress tolerance, or high yield under optimal conditions can inform crossing strategies to maximize genetic gain and develop hybrids suitable for both marginal and favorable environments.

### 4.6 Integrated genetic diversity, heterotic structure, and cluster-based selection for enhanced hybrid breeding efficiency

The distribution of doubled haploid (DH)-derived testcrosses across clusters highlights the importance of both within- and between-heterotic group diversity in maize breeding. The dispersion of lines from the same biparental populations into multiple clusters indicates substantial within-population genetic variation, confirming that DH populations can capture recombination-driven diversity and generate broad phenotypic variation. The finding agrees with previous reports [32,36,60,61].

Genotypes from the same heterotic group were distributed across different clusters, suggesting considerable within-group diversity. This is consistent with reports that recombination, introgression, and selection history can create sub-structuring within heterotic groups [36,62,63]. The presence of contrasting genetic backgrounds among clusters indicates that testcrossing DH lines with a tester from the opposite heterotic group remains effective for capturing combining ability and heterosis [32,62]. Hybrids derived from inter-group crosses are often associated with superior grain yield and heterotic expression compared to within-group crosses.

The clustering also reflects adaptive differentiation, with specific groups enriched for drought tolerance, high yield, or introgressed backgrounds. Similar results have been reported in multi-trait clustering studies [32,36]. Integrating genomic and phenotypic data further enhances the identification of adapted and high-performing germplasm [61]. By considering multiple agronomic and adaptive traits simultaneously, cluster-based selection improves the identification of well-balanced genotypes and enhances selection efficiency under complex environments [32,60].

The relationship between genetic distance and heterosis remains complex. While several studies confirmed a positive association particularly for inter-heterotic group crosses, some reports indicates that genetic distance alone is not a reliable predictor [64,65]. The present findings suggested that genetic divergence among DH lines exists but is not fully captured by conventional heterotic grouping. The wide dispersion of lines across clusters indicates complex and admixed diversity. Therefore, even without clear heterotic group separation, sufficient divergence between DH lines and testers can still generate heterotic responses. Furthermore, it has been confirmed that the DH lines derived from biparental populations harbor substantial genetic variation, and that testcrossing with an opposite heterotic group tester remains an effective strategy to exploit this diversity. Integrating DH technology, heterotic grouping, and multi-trait clustering provides a robust framework for improving selection accuracy, maximizing heterosis, and accelerating genetic gain in maize breeding, particularly under drought-prone conditions.

### 4.7 Association of traits

In this study, which shows that grain yield is controlled by a coordinated network of flowering, architectural, and ear-related traits, agrees with both earlier and recent findings. Grain yield is a complex trait influenced by multiple interrelated components rather than by single traits acting independently [66]. The positive associations of grain yield with ears per plant, plant height, and ear height support earlier reports that these traits enhance sink capacity and assimilate partitioning in maize [4,67]. These traits have therefore been widely recommended as useful indirect selection criteria for yield improvement. In contrast, negative relationships between grain yield and flowering delay traits, particularly the anthesis–silking interval (ASI), are well documented. Extended ASI has been associated with poor reproductive success and reduced grain yield, especially under drought and low-nitrogen conditions [4,68]. Recent studies confirm that genotypes

with shorter ASI show greater yield stability across environments [66,69,70]. The central role of ASI identified in this study is supported by physiological and molecular evidence. Reduced ASI improves pollen–silk synchrony and reproductive efficiency, leading to higher grain yield under both stress and non-stress conditions [71]. Overall, these findings reinforce that simultaneous selection for short ASI, favorable plant architecture, and ear productivity is more effective than selection based on grain yield alone. This integrated trait-based approach provides a practical and sustainable framework for yield improvement under variable and stress-prone environments.

## 5 Conclusions

This study confirms the hypothesis that doubled haploid (DH) lines from biparental maize populations retain sufficient genetic variability. This variability generates exploitable phenotypic diversity and supports adaptation across environments. Genotypes were widely distributed across clusters, indicating strong adaptive differentiation. Heterotic grouping alone did not fully explain genetic diversity or performance. Multivariate approaches, such as PCA and clustering, more effectively captured functional variation linked to adaptation. Clusters acted as adaptive breeding units by integrating genetic background with environmental response. Strong genotype × environment interaction and limited stability of superior genotypes highlighted the need for environment-specific selection. Multi-trait selection is also essential due to the complex control of grain yield. Scientifically, the study demonstrates how DH-derived variability can be structured and exploited using multivariate tools. For breeding, it provides a practical framework for parent selection and hybrid development. Integrating DH technology, heterotic grouping, and cluster-based approaches enhances heterosis and accelerates genetic gain. This strategy supports the development of high-yielding and drought-resilient maize hybrids for variable and stress-prone environments.

### Key Message

- Doubled haploid lines created useful diversity within the same heterotic group
- Some hybrids strongly outperformed commercial checks under both optimal and drought conditions
- Strong genotype × environment interaction affected stability
- Cluster-based selection can improve drought tolerance and yield stability

## Acknowledgments

The authors are grateful to the International Maize and Wheat Improvement Center (CIMMYT) scientists and technicians who generated the germplasm, and highly appreciate the technical support received from the staff members affiliated to CIMMYT maize research station in Kiboko, Kenya.

## Author contributions

**Conceptualization:** Goshime Muluneh Mekasha, Zerihun Demrew Yigezu, Adefris Teklewold, Manje Gowda, Juan Burgueño, Yoseph Beyene.

**Data curation:** Goshime Muluneh Mekasha, Manje Gowda.

**Formal analysis:** Goshime Muluneh Mekasha.

**Investigation:** Adefris Teklewold, Yoseph Beyene.

**Methodology:** Goshime Muluneh Mekasha, Adefris Teklewold, Manje Gowda, Juan Burgueño, Yoseph Beyene.

**Resources:** Yoseph Beyene.

**Supervision:** Zerihun Demrew Yigezu, Adefris Teklewold, Manje Gowda, Juan Burgueño, Yoseph Beyene.

**Visualization:** Goshime Muluneh Mekasha.

**Writing – original draft:** Goshime Muluneh Mekasha.

**Writing – review & editing:** Goshime Muluneh Mekasha, Zerihun Demrew Yigezu, Adefris Teklewold, Manje Gowda, Juan Burgueño.

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
