## [Decision Letter · Decision Letter 0]

6 Apr 2026

PONE-D-26-121031 Performance of Doubled Haploid Maize (Zea mays L.) Testcross Hybrids Across Optimal and 2 Drought-Stressed EnvironmentsPLOS One

Dear Dr. Mekasha,

Thank you for submitting your manuscript to PLOS ONE. After careful consideration, we feel that it has merit but does not fully meet PLOS ONE’s publication criteria as it currently stands. Therefore, we invite you to submit a revised version of the manuscript that addresses the points raised during the review process.

**ACADEMIC EDITOR:**

Dear Authors,

We have now received the reviewers’ comments on your manuscript. The reviewers find the study to be of interest and believe it has the potential for publication. However, they have also indicated that the manuscript requires substantial revision before it can be considered further.

We kindly ask you to revise the manuscript in accordance with the reviewers’ comments and submit a revised version along with a detailed rebuttal letter addressing each point raised.

We look forward to receiving your revised submission.

Thank you.

We look forward to receiving your revised manuscript.

Kind regards,

C Anilkumar, Ph.D.

Academic Editor

PLOS One

Journal Requirements:

“Bill and Melinda Gates Foundation (B&MGF), and the United States Agency for International Development (USAID) through the Stress Tolerant Maize for Africa (STMA, B & MGF Grant # OPP1134248)”

“no”

5. Please be informed that funding information should not appear in the Acknowledgments section or other areas of your manuscript. We will only publish funding information present in the Funding Statement section of the online submission form. Please remove any funding-related text from the manuscript.

Additional Editor Comments (if provided):

Dear Authors,

We have now received the reviewers’ comments on your manuscript. The reviewers find the study to be of interest and believe it has the potential for publication. However, they have also indicated that the manuscript requires substantial revision before it can be considered further.

We kindly ask you to revise the manuscript in accordance with the reviewers’ comments and submit a revised version along with a detailed rebuttal letter addressing each point raised.

We look forward to receiving your revised submission.

Thank you.

Reviewers' comments:

Reviewer's Responses to Questions

**Comments to the Author**

1. Is the manuscript technically sound, and do the data support the conclusions?

Reviewer #1: Yes

Reviewer #2: Yes

2. Has the statistical analysis been performed appropriately and rigorously? 

Reviewer #1: Yes

Reviewer #2: Yes

3. Have the authors made all data underlying the findings in their manuscript fully available?

Reviewer #1: Yes

Reviewer #2: Yes

4. Is the manuscript presented in an intelligible fashion and written in standard English?

Reviewer #1: Yes

Reviewer #2: Yes

5. Review Comments to the Author

Reviewer #1: Abstract and Introduction

• Explicitly state the selection pressure applied (e.g., top 1.1% of 855 hybrids) to highlight the rigor of identifying the ten best performers.

• Clarify the specific heterotic groups (e.g., Group A vs. Group B) mentioned in the hypothesis to provide immediate context for maize breeders.

• In the introduction, move beyond general drought statistics to specifically address the increasing frequency of mid-season "managed drought" scenarios common in Sub-Saharan Africa.

• Briefly mention the time-saving advantage of Doubled Haploid technology compared to traditional inbreeding to underscore the study's relevance to "rapidly developing germplasm".

Materials and Methods

• Explain the "common checks" strategy used to link the 17 trial sets more clearly, ensuring the reader understands how 855 hybrids were compared fairly.

• Quantify "optimal" vs. "managed drought" in terms of millimeters of water applied or soil moisture depletion levels.

• Ensure all software versions (DeltaGen, R-software 4.4.1) are consistently cited with their respective developers or URLs.

Results

• Explicitly discuss the 37.73% Coefficient of Variation (CV) for the Kiboko_DT trial; explain if this is typical for high-intensity drought stress in this region.

• Provide a summary sentence noting that heritability for phenological traits (AD, PH, EH) was generally higher than for grain yield (GY) across all environments.

• Explain the statistical weighting used for Pop 17 and 18, given they were represented by only two and four genotypes respectively.

• Clearly state which traits (e.g., GY, EPP) were the primary drivers of PC1 and PC2 variation based on their vector lengths in Figure 6.

• Emphasize that only seven genotypes (5.5%) were common across all environments to reinforce the "strong genotype x environment interaction" finding.

Discussion

• Compare your 161% yield advantage finding with other tropical DH studies to show if your genetic gains are standard or exceptional.

• Expand the discussion on why the short ASI in Cluster 3 (1.02–1.35 days) is the "critical" physiological driver for drought resilience.

• Address how the 23.4% "unknown" background genotypes in Cluster 1 might contribute to the observed functional diversity.

• Discuss why residual variance for ASI was 38% higher in well-watered conditions compared to stress.

• Explicitly recommend "cluster-based selection" as a superior alternative to single-trait selection based on your multivariate results.

Reviewer #2: The manuscript presents a robust set of experimental data and addresses a relevant topic in maize breeding under water stress conditions. The approach involving doubled haploid lines and multi-environment evaluation is appropriate and timely. However, the study exhibits conceptual inconsistencies, methodological gaps, and weaknesses in the articulation between hypothesis, results, and discussion, which must be addressed to strengthen its scientific contribution.

Abstract

The abstract is overly long and could be reduced without loss of content. It is recommended to make it more concise, prioritizing objectives, methodological approach, key results, and conclusions, while avoiding excessive detail.

Introduction

The introduction contains an extensive review that could be shortened without compromising the study’s contextualization.

There is no clear need to address marker-assisted selection, as this topic is not developed throughout the manuscript.

Additionally, the abstract presents an explicit hypothesis; however, this hypothesis is neither revisited nor properly justified in the introduction. It is recommended that the authors clearly incorporate this hypothesis into the introduction, establishing a logical link between the scientific gap, study objectives, and experimental approach.

Materials and Methods

The methodological description requires greater clarity and detail in several aspects:

The statement that “each of the 855 DH lines was crossed with a single-cross tester from the opposite heterotic group” introduces a conceptual inconsistency with the study hypothesis, which suggests the exploration of variability within the same genetic background. The use of a tester from an opposite heterotic group may introduce substantial genetic divergence and therefore requires clarification. The authors should explicitly explain this strategy and its relationship to the proposed hypothesis.

Table 1 lacks an adequate explanatory footnote, including full descriptions of abbreviations and presented information. This issue also applies to several other tables and figures. For example, what does “HG” refer to?

The characterization of experimental environments is insufficient. Terms such as “well-watered” and “drought stress” require clear operational definitions. It is recommended to include:

irrigation management criteria;

monitoring of drought stress;

occurrence and amount of rainfall during the experimental period;

indicators confirming that drought stress was effectively imposed.

These details are essential for proper interpretation of the results.

Discussion

The discussion section presents several important weaknesses:

Multiple statements are not adequately supported by bibliographic references. For example, the claim of higher heritability under drought stress should be properly referenced. A careful revision is recommended to ensure consistent bibliographic support.

The central hypothesis of the study is not revisited or discussed, which compromises the overall coherence of the manuscript. The discussion should be reorganized to explicitly address the proposed hypothesis.

A relevant conceptual issue was not addressed: the apparent lack of substantial genetic distance among DH lines contrasted with the use of a tester from an opposite heterotic group. Considering that heterosis is associated with genetic divergence and heterotic group structure, this point should be explicitly discussed.

The discussion is largely descriptive, lacking deeper interpretation and stronger integration with the literature.

Conclusion

The conclusion largely repeats the results without effectively synthesizing the implications of the study.

There is no clear closure of the proposed hypothesis, indicating inconsistency between the introduction, development, and conclusion. The section should be reformulated to:

explicitly address the hypothesis;

highlight the scientific contributions;

indicate implications for plant breeding.

6. PLOS authors have the option to publish the peer review history of their article (what does this mean?). If published, this will include your full peer review and any attached files.

Reviewer #1: No

Reviewer #2: No

---

## [Author Response · Author response to Decision Letter 1]

5 May 2026

We have carefully revised the manuscript following the reviewers’ and editor’s comments. A detailed, point-by-point response to all comments has been provided in the attached response document, and all changes have been incorporated and highlighted in the revised manuscript.We have carefully revised the manuscript following the reviewers’ and editor’s comments. A detailed, point-by-point response to all comments has been provided in the attached response document, and all changes have been incorporated and highlighted in the revised manuscript.

---

## [Decision Letter · Decision Letter 1]

19 May 2026

Performance of Doubled Haploid Maize (Zea mays L.) Testcross Hybrids Under Optimal and Drought-Stressed Environments

PONE-D-26-12103R1

Dear Dr. Mekasha,

We’re pleased to inform you that your manuscript has been judged scientifically suitable for publication and will be formally accepted for publication once it meets all outstanding technical requirements.

Kind regards,

C Anilkumar, Ph.D.

Academic Editor

PLOS One

Additional Editor Comments (optional):

Dear authors,

Thank you for revising the manuscript in accordance with the reviewer comments. Both reviewers are satisfied with the revisions and have recommended the manuscript for acceptance. The article has been significantly improved in terms of technical rigor and overall presentation. I recommend the manuscript for publication.

Reviewers' comments:

Reviewer's Responses to Questions

**Comments to the Author**

1. If the authors have adequately addressed your comments raised in a previous round of review and you feel that this manuscript is now acceptable for publication, you may indicate that here to bypass the “Comments to the Author” section, enter your conflict of interest statement in the “Confidential to Editor” section, and submit your "Accept" recommendation.

Reviewer #1: All comments have been addressed

Reviewer #2: All comments have been addressed

2. Is the manuscript technically sound, and do the data support the conclusions?

Reviewer #1: Yes

Reviewer #2: Yes

3. Has the statistical analysis been performed appropriately and rigorously? 

Reviewer #1: Yes

Reviewer #2: Yes

4. Have the authors made all data underlying the findings in their manuscript fully available?

Reviewer #1: Yes

Reviewer #2: Yes

5. Is the manuscript presented in an intelligible fashion and written in standard English?

Reviewer #1: Yes

Reviewer #2: Yes

6. Review Comments to the Author

Reviewer #1: The authors have submitted a revised version of the manuscript and provided a detailed response to the reviewers' concerns from the first round of review. The authors have made commendable efforts to improve the clarity, synthesis, and scientific rigor of the paper, particularly regarding the framing of their hypothesis and the synthesis of their conclusions.

Assessment of Revisions

• The authors have expanded the discussion section and successfully incorporated appropriate, up-to-date literature to better compare and support their findings regarding DH line performance under drought stress.

• In the previous round, it was noted that the conclusion largely repeated the results. The authors have adequately addressed this by revising the section (lines 803-815) to focus on synthesizing the broader implications of the study, rather than merely summarizing data.

• The authors successfully reformulated the text to explicitly address the proposed hypothesis—specifically concerning whether DH lines derived from biparental populations within the same heterotic group can generate exploitable phenotypic diversity. They have also better highlighted the scientific contributions and practical implications for plant breeding programs.

• The authors satisfactorily addressed the additional journal requirement regarding field site permits, clarifying that the work falls under the CIMMYT mandate for developing and evaluating new lines across Sub-Saharan Africa (lines 104-107), requiring no specific external permissions.

Reviewer #2: The authors addressed the suggested revisions. I believe the authors carefully considered the reviewer’s comments and satisfactorily addressed the concerns raised. Therefore, I recommend acceptance of the manuscript.

7. PLOS authors have the option to publish the peer review history of their article (what does this mean?). If published, this will include your full peer review and any attached files.

Reviewer #1: No

Reviewer #2: No

---

## [Editor Report · Acceptance letter]

PONE-D-26-12103R1

PLOS One

Dear Dr. Mekasha,

I'm pleased to inform you that your manuscript has been deemed suitable for publication in PLOS One. Congratulations! Your manuscript is now being handed over to our production team.

Kind regards,

on behalf of

Dr. C Anilkumar

Academic Editor

PLOS One